# Systematic Mendelian randomization using the human plasma proteome to discover potential therapeutic targets for stroke

Lingyan Chen [1,2], James E. Peters [1,3], Bram Prins [1], Elodie Persyn [1,4,5], Matthew Traylor[2,6], Praveen Surendran[1,7], Savita Karthikeyan[1], Ekaterina Yonova-Doing[1,2], Emanuele Di Angelantonio[1,8,9,10,11], David J. Roberts[9,12,13], Nicholas A. Watkins[14], Willem H. Ouwehand [8,14,15,16], John Danesh[1,8,9,10,17], Cathryn M. Lewis [4,18], Paola G. Bronson [19], Hugh S. Markus[20], Stephen Burgess [1,8,21], Adam S. Butterworth [1,8,9,10] & Joanna M. M. Howson [1,2] ✉

Stroke is the second leading cause of death with substantial unmet therapeutic needs. To identify potential stroke therapeutic targets, we estimate the causal effects of 308 plasma proteins on stroke outcomes in a two-sample Mendelian randomization framework and assess mediation effects by stroke risk factors. We find associations between genetically predicted plasma levels of six proteins and stroke ($P \leq 1.62 \times 10^{-4}$). The genetic associations with stroke colocalize (Posterior Probability >0.7) with the genetic associations of four proteins (TFPI, TMPRSS5, CD6, CD40). Mendelian randomization supports atrial fibrillation, body mass index, smoking, blood pressure, white matter hyperintensities and type 2 diabetes as stroke risk factors ($P \leq 0.0071$). Body mass index, white matter hyperintensity and atrial fibrillation appear to mediate the TFPI, IL6RA, TMPRSS5 associations with stroke. Furthermore, thirty-six proteins are associated with one or more of these risk factors using Mendelian randomization. Our results highlight causal pathways and potential therapeutic targets for stroke.

Stroke is the second leading cause of death worldwide, estimated to cause ~6.5 million deaths annually, and is the leading cause of long-term disability, with a growing burden on global health[1]. Therefore, there is a need for new and improved treatments and prevention strategies for stroke. While conventional risk factors, such as hypertension[2], account for ~50% of stroke risk, there remains a need to identify new risk factors, biomarkers and therapies for stroke[3]. In 2017, ~75% of FDA-approved drugs were targeted at human proteins[4]. Plasma proteins play a central role in a range of biological processes frequently dysregulated in diseases[5–8], and represent a major source of therapeutic targets for many indications[4,9,10]. In particular, plasma proteins are particularly relevant for circulatory diseases such as stroke as they are in physical

contact with the blood vessels (compared to tissue-specific diseases, e.g. inflammatory bowel disease[11]).

Genome-wide association studies (GWAS) of plasma protein levels have identified genetic variants that are associated with proteins, usually referred to as 'protein quantitative trait loci (pQTLs)'[12–17], offering an opportunity to test the causal effect of potential drug targets on the human disease phenome using Mendelian randomisation (MR)[18,19]. Briefly, MR can be thought of as nature's randomised trial, by capitalising on the random allocation of genetic variants at conception to separate individuals into subgroups (one equivalent to placebo and the other to intervention in a randomized control trial, RCT) and so allows testing of the potential causal association of risk

---

factors (e.g. plasma proteins) with disease outcomes (*e.g.* stroke) as confounders should also be randomised[20].

Here, we perform a two-sample MR to estimate the causal effects of plasma proteins on stroke, where we derived genetic instrumental variables (IV) of 308 circulating plasma proteins from 4994 participants[21] and obtained genetic associations of stroke subtypes (any stroke (AS), any ischaemic stroke (IS), large artery stroke (LAS), cardioembolic stroke (CES) and small vessel stroke (SVS)) from the MEGASTROKE GWAS[22]. Then, to verify the robustness of the proteins' instrumental variables, we perform colocalization analyses. We evaluate the causal relationship of plasma proteins on stroke risk factors and assess the potential safety effects of targeting the proteins for stroke therapy by performing a phenome-wide MR in UK Biobank GWASs[23].

## Results

### Identification of stroke-associated proteins

Three hundred and eight plasma proteins were tested for causal relationships with stroke outcomes (Fig. 1 and Supplementary Data 1, 2). As *cis*-pQTLs were considered to have a more direct and specific biological effect on the protein (compared to *trans*-pQTLs)[24], we first performed MR analyses using only *cis*-pQTLs as instrumental variables and identified six putatively causal proteins with at least one stroke outcome ($P \leq 1.62 \times 10^{-4} = 0.05/308$ proteins; Figs. 2, 3 and Supplementary Fig. 1): TFPI (tissue factor pathway inhibitor), TMPRSS5 (Transmembrane Serine Protease 5), CD40 (B Cell Surface Antigen CD40), MMP12 (Matrix Metallopeptidase 12), IL6RA (Interleukin 6 Receptor) and CD6 (T-Cell Differentiation Antigen CD6). TFPI, CD40, IL6RA and MMP12 were associated with a lower risk of any stroke and any ischaemic stroke, while TMPRSS5 and CD6 was associated with a higher risk of any stroke. Among the ischaemic stroke subtypes, genetic predisposition to upregulated TMPRSS5 was associated with higher risk of any ischaemic stroke (OR per-1-SD higher plasma protein level [95% CI] = 1.059[1.038, 1.08]; $P = 1.36 \times 10^{-8}$) and cardioembolic stroke (OR[95%CI] = 1.089[1.045, 1.134]; $P = 5.33 \times 10^{-5}$). Higher genetically predicted levels of both MMP12 (OR[95%CI] = 0.793[0.73, 0.861]; $P = 3.53 \times 10^{-8}$) and CD40 (OR[95% CI] = 0.795[0.723, 0.874]; $P = 2.09 \times 10^{-6}$) were associated with lower risk of large artery stroke. Higher genetically predicted soluble IL6RA (and lower IL6R signalling[25]) was associated with lower risk of small vessel stroke (OR[95% CI] = 0.939[0.909, 0.970]; $P = 1.60 \times 10^{-4}$).

We extended the MR analyses to include *trans*-pQTLs as instrumental variables and identified nine additional proteins associated with at least one stroke outcome ($P \leq 1.62 \times 10^{-4}$; Supplementary Data 3). However, seven proteins (VSIG2, EPHB4, Gal4, ICAM2, LIFR, SELE and vWF), included instrumental variables from the *ABO* locus, which is well known to have pleiotropic effects. We note that the ABO protein has previously been identified as a genetic risk factor for stroke[26]. Interestingly, both Bone Morphogenetic Protein 6 (BMP6) and Growth Differentiation Factor 2 (GDF2, also known as BMP9) were instrumented by *trans*-pQTLs located in the genetic regions of *KNG1* (Kininogen 1) and *F11* (Coagulation Factor XI). Both genes are essential for blood coagulation and the latter has previously been reported to be a causal risk factor for stroke[27]. GDF2 has also been found to have a causal role in pulmonary artery hypertension (PAH)[28]. We, therefore, focused further analyses on the proteins with *cis*-pQTLs only (i.e. TFPI, TMPRSS5, CD40, MMP12, IL6RA, CD6), as these associations with stroke are unlikely to be due to pleiotropy.

Results of sensitivity analyses confirmed the robustness of the primary MR analyses. There was no evidence for heterogeneity in the association of any of the six proteins in Supplementary Data 3 as measured by Cochran Q statistics ($P_{Q-stat} > 0.05$), and no indication that the instrumental variables had horizontal pleiotropy as assessed by MR-Egger intercept ($P_{Egger-Intercept} > 0.05$) or MR-PRESSO global pleiotropy test ($P_{GlobalTest} > 0.05$). All MR causal effect estimates

adjusting for correlation of IVs were consistent with the primary analyses (Supplementary Data 4). Moreover, MR causal estimates using IVs derived from conditionally independent variants and credible sets of variants from fine-mapping showed consistent results (Supplementary Data 5, 6). There was no evidence of reverse causations (Supplementary Data 7).

### Shared genetic associations with protein levels and risk of stroke

We formally tested whether the associations of the variant with the protein levels used as IVs and the stroke outcome are shared for the six proteins using statistical colocalization analysis. We applied a Bayesian algorithm, Hypothesis Prioritisation in multi-trait Colocalization (HyPrColoc), which allows for the assessment of colocalization across multiple complex traits simultaneously (Methods), to test whether the protein associations and stroke associations are shared. The association of the genetic variants selected as instrumental variables for four proteins (TFPI, TMPRSS5, CD40 and CD6) colocalized with the stroke associations (posterior probability (PP) ≥0.7) (Supplementary Data 8 and Supplementary Fig. 2) i.e. the associations in these regions were likely due to the same underlying causal variants. The colocalization suggested the genetic variants associated with TFPI (pQTLs) were due to the same genetic variants underlying the association with any stroke. Similarly, CD6 pQTLs colocalized with any stroke genetic associations; CD40 pQTLs colocalized with the genetic associations for any stroke, any ischaemic stroke and large artery stroke; TMPRSS5 pQTLs colocalized with any stroke, any ischaemic stroke and cardioembolic stroke genetic associations. Notably, we found for TFPI, CD40 and CD6 that >80% of the posterior probability of colocalization of the primary genetic association with stroke and the respective protein levels were explained by a single variant (rs67492154, rs4810485 and rs2074227 for TFPI, CD40 and CD6, respectively). The colocalization evidence at MMP12, was less strong than with the other proteins, with colocalization PP >0.6 and there was no colocalization evidence for IL6RA with stroke, which could be due to violation of the single causal variant assumption of the HyprColoc method.

### Identification of likely causal stroke risk factors

To understand potential causal mechanisms between plasma proteins and stroke, we conducted two-step mediation MR analyses for conventional stroke risk factors. First, we performed two-sample MR analyses to characterise the causal relationship of the stroke risk factors with all stroke outcomes. Second, we assessed the causal effects of the proteins on the highlighted risk factors.

For each of the seven stroke risk factors we considered (*i.e.*, blood pressure (BP), atrial fibrillation (AF), type 2 diabetes (T2D), white matter hyperintensity (WMH), body mass index (BMI), smoking behaviours and alcohol consumption), instrumental variables were derived from published GWAS summary statistics restricted to European populations (Table 1 and Supplementary Data 9). AF, T2D, smoking, increased systolic BP, diastolic BP, pulse pressure, WMH and BMI increased the risk of any stroke ($P \leq 0.05/7 = 0.007$, Bonferroni-adjusted for seven risk factors; Fig. 4; Supplementary Data 10 and Supplementary Fig. 3). As expected, systolic BP exhibited the strongest effect of all the risk factors on any ischaemic stroke and LAS (OR per-1-SD [95% CI] = 1.68[1.57, 1.80] and 2.58[2.21, 3.01], respectively) and AF had a positive association with CES (OR[95% CI]: 2.04[1.92, 2.16]; $P = 2.72 \times 10^{-125}$). WMH increased risk of any stroke and SVS (1-SD increased in WMH was associated with 49% higher odds for SVS (OR[95% CI] = 1.49[1.17, 1.9]; $P = 0.00147$). Both genetically determined higher T2D risk and smoking initiation were associated with increased risk of LAS and SVS; and genetically determined higher BMI was associated with a higher risk of LAS. No association was observed between alcohol consumption with any of the stroke outcomes ($P > 0.05$).

## 1. Identify candidate stroke-associated proteins

GWAS of 354 Plasma Proteins
[≤ 4,994 INTERVAL participants]

↓

selection of IVs [$P ≤ 5.0 × 10^{-8}$ , $r^2 ≤ 0.10$]

↓

**MR: Proteome → Stroke outcomes #**

↓

| cis-pQTLs only | cis + trans-pQTLs |
|---|---|

Sensitivity analyses

↓

| **6 stroke-associated proteins** | **15 stroke-associated proteins** |
|---|---|

↓

Colocalization analysis

## 2. Identify stroke risk factors

Identify stroke risk factors from literature review

↓

Identify largest GWAS for risk factors & Select IVs

↓

**MR: Risk factors → Stroke outcomes**

↓

**6 risk factors are likely causal for Stroke**
- SBP, DBP, PP
- AF
- WMH
- T2D
- BMI
- Smoking

## 3. Identify stroke risk factors associated proteins

| **Exposure:** 234 proteins with cis-pQTLs | **Outcome:** 6 stroke risk factors |
|---|---|

↓

**MR: Proteome → Causal risk factors**

↓

cis-pQTLs only

Sensitivity analyses

↓

**39 proteins associated with at least one stroke risk factor(s)**

↓

Colocalization analysis

## 4. Perform mediation analysis

**3** proteins associated with stroke outcomes and risk factors

↓

**Mediation analysis: Two-step MR**

↓

SNP(s)

Stroke Risk factor (Mediator)

pQTLs → Protein → Stroke Outcomes

## 5. Perform Phenome-wide MR

| **Exposure:** 6 stroke-associated proteins | **Outcome:** 784 phenotypes from UKBB GWAS |
|---|---|

↓

**PheMR: 6 proteins → Phenome-wide outcomes**

↓

cis-pQTLs only

Sensitivity analyses

↓

Evaluation of side-effects & additional indications

↓

Colocalization analysis

**Fig. 1 | Overview of this MR study.** Four O-link panels were used to measure plasma proteins in a subset of ~5000 samples from the INTERVAL study. Genetic variants associated with plasma protein levels were identified based on results from their corresponding GWAS. These genetic variants were then used as proxies for the protein level to test their relationship with stroke using data from the MEGA-STROKE consortium for stroke outcomes (Primary MR), and with conventional stroke risk factors that were causally associated with stroke (Secondary MR). Colocalization analyses were performed to test the shared genetic associations of protein level, stroke outcomes and risk factors. Mediation analyses by two-step MR were performed for proteins that were potentially causally associated with both risk factors and stroke outcomes. We also tested the relationships of the potentially causal stroke proteins with 784 phenotypes in the UK Biobank to test a broad spectrum of potential effects of hypothetical therapeutic agents for stroke. #Stroke outcomes: any stroke; any ischaemic stroke; large artery stroke; cardioembolic stroke; small vessel stroke.

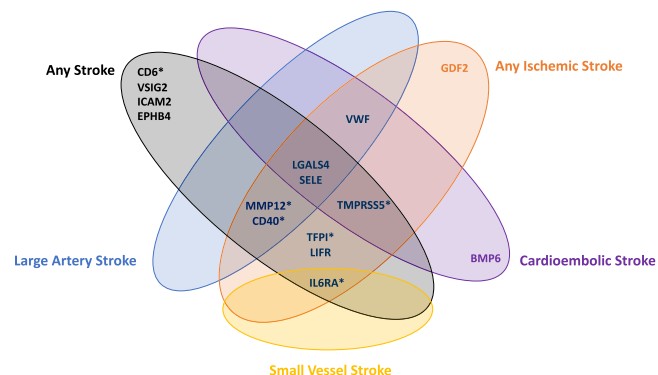

**Fig. 2 | Venn diagram of identified potential causal proteins for stroke subtypes.** * Indicates the according to protein is instrumented by *cis*-pQTLs only. These six proteins are taken for further analyses.

### Identification of stroke risk factors associated proteins

We performed MR of all 308 plasma proteins with the six highlighted stroke risk factors (excluding alcohol consumption which was not associated with increased stroke risk in the above MR analyses). After multiple testing correction, 39 proteins instrumented with *cis*-pQTLs were associated with at least one stroke risk factor ($P ≤ 1.62 × 10^{-4}$): five with systolic BP; seven with diastolic BP; seven with pulse pressure; six with AF; four with T2D; nine with BMI; three with WMH; and eight with smoking. There was no evidence of horizontal pleiotropy, and sensitivity analyses yielded consistent causal effect estimates (Supplementary Data 11).

Among the six stroke-associated proteins, three proteins were found to be associated with one or more of the stroke risk factors (Fig. 5 and Supplementary Fig. 4). Of note, we found that genetically predicted higher TFPI level was associated with lower WMH and lower BMI (a 0.06 SD lower WMH β[95% CI] = −0.06[−0.08, −0.04]; $P = 7.15 × 10^{-10}$

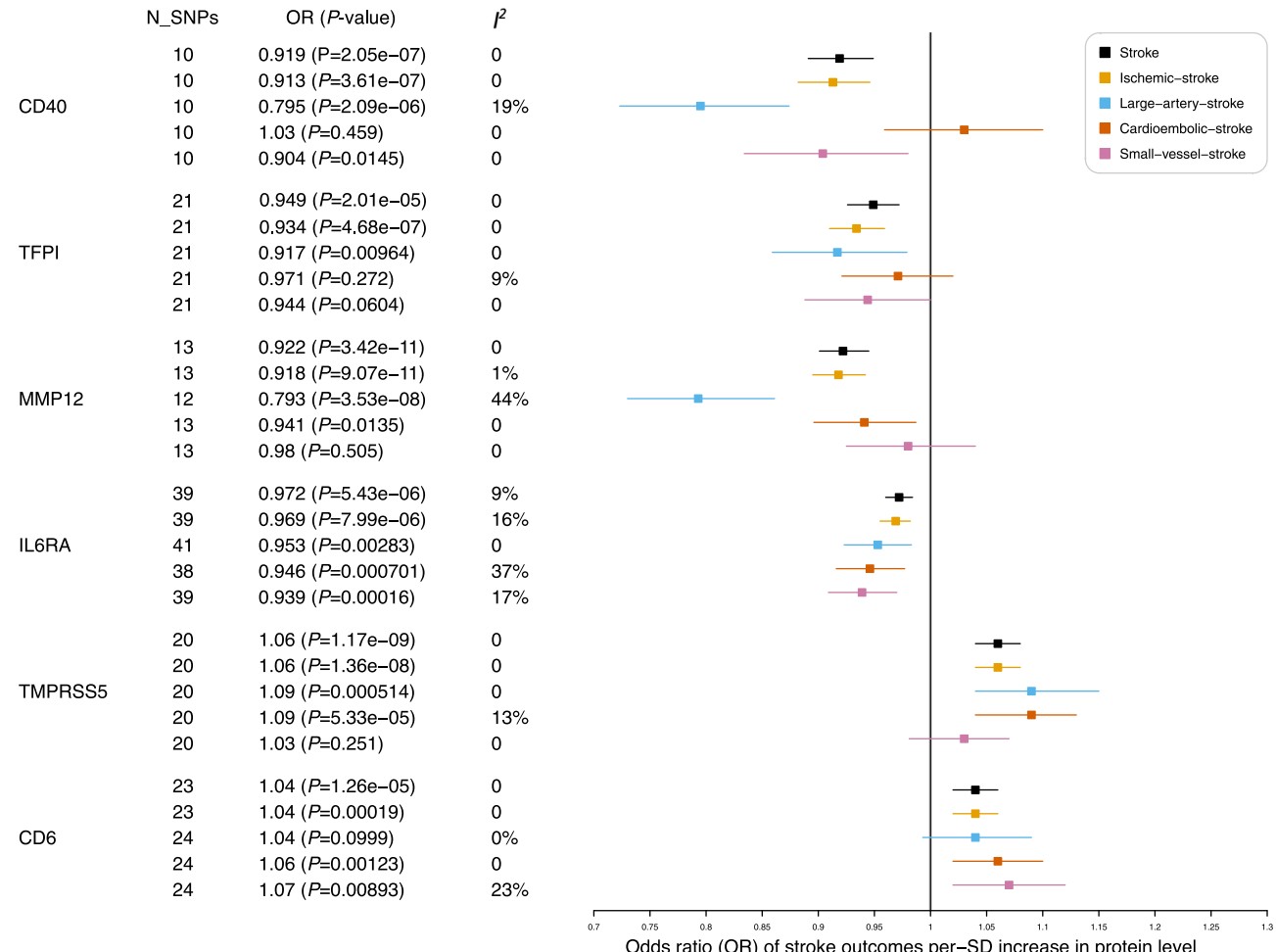

**Fig. 3 | Effects of six potential causal proteins on stroke outcomes.** MR analyses of the effect of proteins on stroke outcomes. The squares are the causal estimates on the OR scale, and the whiskers represent the 95% confidence intervals for these ORs. N_SNPs: number of SNPs used for the estimation of the causal effects in this plot. *P* values were determined from the inverse-variance-weighted two-sample MR method. Statistical heterogeneity was assessed using the $I^2$ statistic. OR odds ratio, CD40 B cell surface antigen CD40, TFPI tissue factor pathway inhibitor, MMP12 matrix metallopeptidase 12, IL6RA interleukin 6 receptor subunit alpha, TMPRSS5 transmembrane serine protease 5, CD6 T-cell differentiation antigen CD6.

and a 0.013 SD lower BMI β[95% CI] = −0.013[−0.019, −0.007]; $P = 3.56 \times 10^{-5}$ per-SD higher TFPI; Supplementary Data 12). Genetically determined higher TMPRSS5 levels were associated with higher risk of AF (OR[95% CI]: 1.03[1.016, 1.045]; $P = 2.15 \times 10^{-5}$). Genetically higher IL6RA levels were associated with a 4.1% lower risk of AF (OR[95% CI]: 0.96[0.95, 0.97]; $P = 2.55 \times 10^{-18}$). All the effect directions of these associations of proteins with risk factors were consistent with those of the proteins with stroke, indicating that these risk factors may be potential mediators of the protein-stroke associations.

Among the 39 proteins that were associated with at least one stroke risk factor, 36 were found to be associated with the risk factors but not stroke outcome (Supplementary Data 11). For example, genetically determined Fibroblast Growth Factor 5 (FGF5) level was associated with a higher risk of AF (OR = 1.056 per-SD higher FGF5); each SD higher genetically determined Glypican 5 (GPC5) was associated with a higher risk of T2D (OR = 1.02); each SD higher in genetically determined scavenger receptor class F member 2 (SCARF2) was associated with a 0.062-SD higher WMH. We found that higher genetically determined Alpha-L-Iduronidase (IDUA) and sialic acid-binding Ig-like lectin 9 (SIGLEC9) were both associated with lower BMI. Higher genetically determined serine protease 27 (PRSS27) was associated with higher SBP, higher DBP and higher PP, while higher genetically determined levels of Neurocan (NCAN) were associated with lower risk of T2D (OR = 0.76) and 0.07-SD lower SBP.

**Mediation effect of proteins on stroke outcomes via risk factors**

To investigate the indirect effect of proteins on stroke outcomes via risk factors, we carried out a mediation analysis using the effect estimates from two-step MR and the total effect from primary MR. This analysis was restricted to three proteins, i.e. TFPI, TMPRSS5 and IL6RA, that showed evidence of an effect in both MRs with risk factors and stroke outcomes. We used the product method to estimate the indirect effect and the delta method to estimate the standard errors (SE) and confidential interval (CI) (Methods). The proportion of mediation effect of TFPI via WMH is about one-fifth (20.8%), while the mediation effect via BMI is modest (3.8%). The indirect effect of TMPRSS5 on the risk of cardioembolic stroke via AF contributes to a quarter of the total effect (24.7%). Similarly, the proportion of mediation effect of IL6RA on stroke via AF is 27.6% (Fig. 6 and Supplementary Data 13).

**Phenome-wide MR (Phe-MR) analysis of stroke-associated proteins**

To assess whether the six stroke-associated proteins have either beneficial or deleterious effects on other indications, we performed a broader MR screen of 784 diseases and traits in the UK Biobank (Supplementary Data 14). Our Phe-MR results can be interpreted as a per-SD increase in genetically determined plasma protein level that leads to a higher or lower odds of a given disease or trait. If the effect direction of the protein on the disease or trait is the same as on stroke,

**Table 1 | Data sources for the Mendelian Randomisation analysis in the current study**

| Phenotype | Sample size # | Imputation reference panel | Ancestry | Source |
|---|---|---|---|---|
| **Olink protein levels** | | | | |
| Inflammation panel (INF1) | 4994 | 1000 Genomes Phase 3 + UK10K | European | INTERVAL study (unpublished data) |
| Cardiovascular panels (CVD2 & 3) | | | | |
| Neurology panel (NEURO) | | | | |
| **Primary outcomes** | | | | |
| Any stroke | 40,585/406,111 | 1000 Genomes phase 1 | European | 17 studies (Malik et al.)[22] |
| Ischaemic stroke | 34,217/406,111 | | | |
| Large artery stroke | 4373/406,111 | | | |
| Cardioembolic stroke | 7193/406,111 | | | |
| Small vessel stroke | 5386/406,111 | | | |
| **Secondary outcomes** | | | | |
| Atrial fibrillation | 60,620/970,216 | HRC | European | 6 Studies (Nielsen, et al.)[76] |
| Type 2 diabetes | 74,124/824,006 | HRC | European | 32 Studies (Mahajan, et al.)[77] |
| Body mass index | 694,649 | HRC | European | GIANT + UK Biobank (Pulit, et al.)[79] |
| Tobacco and alcohol use | | HRC | European | 29 Studies (Liu, et al.)[80] |
| AgeSmk | 341,427 | | | |
| CigDay | 337,334 | | | |
| SmkCes | 547,219 | | | |
| SmkInit | 1,232,091 | | | |
| DrnkWk | 941,280 | | | |
| Blood pressure (BP) | 445,415 | HRC | European | UK Biobank (Surendran, et al.)[75] |
| Systolic BP | | | | |
| Diastolic BP | | | | |
| Pulse pressure | | | | |
| White Matter Hyperintensity | 42,310 | HRC | Trans-ethnic, mainly European | UK Biobank + CHARGE + stroke patients (Persyn et al., 2020) [78] |
| **On-target side-effects evaluation** | | | | |
| 784 Phenotypes | 408,961 | HRC | European | UK Biobank (Zhou, et al.)[91] |

\# Sample size shown as a total number for quantitative traits and Cases/Controls for binary traits.

*UK10K* UK Biobank 10K reference, *HRC* the haplotype reference consortium, *AgeSmk* age of initiation of regular smoking, *CigDay* cigarettes per day, *SmkCes* smoking cessation, *SmkInit* smoking initiation, *DrnkWk* drinks per week.

the effect is considered 'beneficial' and 'deleterious' otherwise. Overall, 34 associations were identified ($P \le 0.05/6/784 = 1.06 \times 10^{-5}$), of which 21 (61.7%) were considered beneficial (Supplementary Data 15).

Notably, genetically higher levels of plasma TFPI were not only associated with a lower risk of stroke, but also a lower risk of other diseases involving the circulatory system (cerebrovascular disease, other disorders of arteries), metabolic traits (hyperlipidemia and hypercholesterolaemia, disorders of lipid metabolism) and digestive system disorders (acute gastritis); however, they were also associated with a higher risk of excessive or frequent menstruation. Genetically higher levels of plasma TMPRSS5 were associated with a higher risk of cardioembolic stroke, as well as protein−calorie malnutrition (metabolic trait) (Fig. 7 and Supplementary Fig. 5). All the associations for CD40, including haemoptysis and abnormal sputum (respiratory system) were considered beneficial. Effects of IL6RA on the risk of diseases on circulatory system disorders (ischaemic heart disease, cardiac dysrhythmias, atrial fibrillation and flutter, coronary atherosclerosis, angina pectoris, abdominal aortic aneurysm) and musculoskeletal disease (other inflammatory spondylopathies) were considered beneficial; but deleterious effects on dermatologic symptoms (e.g. cellulitis and abscess of arm/foot), digestive system (e.g. cholelithiasis) and chronic renal failure [CKD] (Supplementary Fig. 6). Genetically predicted CD6 was associated with alcoholic liver damage and degeneration of intervertebral disc (musculoskeletal system), which were considered deleterious. Summary results of the primary and sensitivity MR analyses for all the 784 phenotypes are provided in Supplementary Data 15.

## Discussion

Based on genetic data for 308 proteins involved in cardiovascular disease, inflammation and neurological processes from ~5000 individuals[21], our study provides robust evidence that six proteins (TFPI, TMPRSS5, CD40, MMP12, IL6RA and CD6) are causally associated with stroke and four of them showed genetic colocalization evidence with stroke outcome(s). We showed that AF, systolic and diastolic BP, BMI, T2D, WMH and smoking were causally associated with risk of any stroke (and some ischaemic stroke subtypes), demonstrating a key role of the risk factors in the pathogenesis of stroke consistent with classical epidemiological data[29–37]. We found the associations of TFPI, IL6RA and TMPRSS5 with stroke were likely to be mediated by one or more of these risk factors. In addition, we showed that 36 additional proteins were causal for these risk factors. Finally, the Phe-MR highlighted additional beneficial indications of therapeutically targeting the six stroke-associated proteins and, importantly, indicated a few potential safety concerns. Although, as many of the phenotypes tested are not independent, the definition of significance used here might be too conservative (Bonferroni-corrected $P$ value adjusted for the number of proteins tested (six) and the total number of phenotypes (784) ($P = 0.05/6/784 = 1.06 \times 10^{-5}$).

Tissue factor pathway inhibitor (TFPI) is primarily secreted by endothelial cells and is an anticoagulant that acts on the clotting cascade[38]. Observational studies showed that lower levels of free TFPI were associated with a higher risk of ischaemic stroke[39] and a higher risk of first and recurrent venous thrombosis[40], while inhibition of TFPI showed to be an effective treatment of bleeding associated with

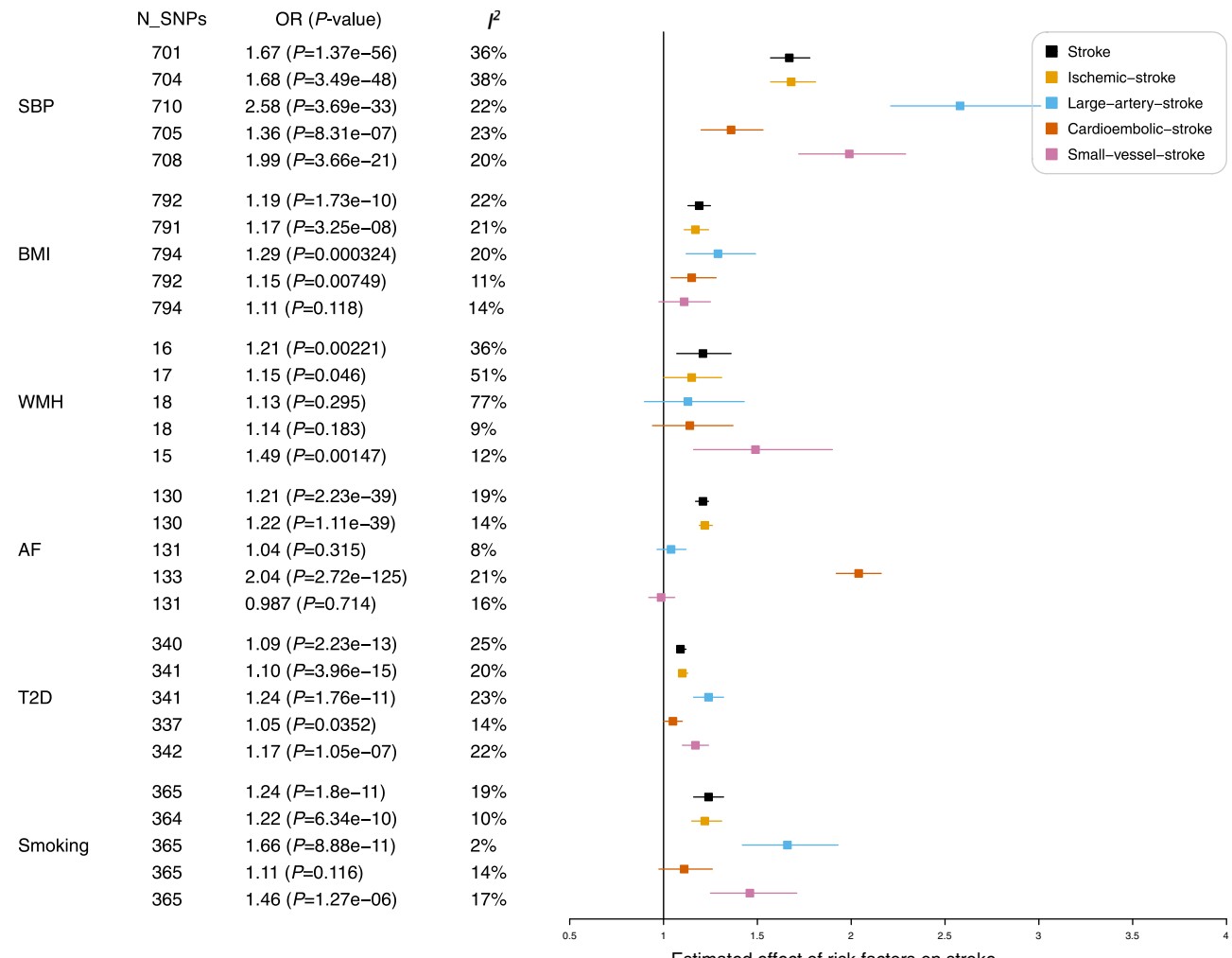

**Fig. 4 | Causal effects of risk factors on stroke outcomes.** MR analyses of the effect of risk factors on stroke outcomes. The squares are the causal estimates on the OR scale, and the whiskers represent the 95% confidence intervals for these ORs. N_SNPs number of SNPs used for the estimation of the causal effects in this plot. *P* values were determined from the inverse-variance-weighted two-sample MR method. Statistical heterogeneity was assessed using the $I^2$ statistic. OR odds ratio, SBP systolic blood pressure, AF atrial fibrillation, WMH white matter hyperintensity, T2D type 2 diabetes, BMI body mass index, Smoking smoking initiation.

haemophilia[41]. Consistent with this, we provided genetic evidence for directionally consistent effects of TFPI on multiple ischaemic traits, such as ischaemic stroke and ischaemic heart disease, and opposite effects on haemorrhagic traits (e.g. gastrointestinal haemorrhage, $P = 5.23 \times 10^{-5}$; excessive or frequent menstruation in females, $P = 2.70 \times 10^{-10}$). We also showed that higher levels of TFPI were associated with lower BMI and WMH (Fig. 5), and lower risk of hyperlipidemia, specifically hypercholesterolaemia (Fig. 7), suggesting that the pathways through which TFPI influences stroke risk might go beyond anticoagulation, e.g. inflammation or atherosclerotic changes. Animal studies[41,42] provide supporting evidence that TFPI has a role in attenuating arterial thrombus formation and atherosclerosis development. Future studies of TFPI in cardiovascular diseases focusing on the role of TFPI activity and different TFPI isoforms in the development of atherogenesis could provide further insights.

TMPRSS5 (transmembrane protease serine 5, also known as Spinesin) is a member of the Type II transmembrane serine protease family (TTSPs)[43]. For example, TMPRSS10 (Corin), one member of the TTSPs, has been reported to be involved in cardiac conduction and myometrial relaxation and contraction pathways in regulating blood pressure and promoting natriuresis, diuresis and vasodilation[44]. Unlike Corin, the function of TMPRSS5 on cardiovascular systems is poorly understood. Human *TMPRSS5* mRNA has been shown to be expressed

in the brain and the protein is predominantly expressed in neurons, in their axons in the spinal cord[45]. A mouse model with mutant TMPRSS5 had reduced proteolytic activity and suggested a role in hearing loss[46]. We were unable to find other studies that implicate TMPRSS5 in cardiovascular disease, both for any ischaemic stroke and cardioembolic stroke, an effect that might be mediated by the risk of atrial fibrillation (Fig. 4). Furthermore, Phe-MR analysis revealed suggestive additional beneficial effects when targeted at TMPRSS5, e.g. reduced risk of Parkinson's disease ($P = 2.15 \times 10^{-5}$) and left bundle branch block ($P = 1.43 \times 10^{-5}$). Taken together, TMPRSS5 represents a potentially promising therapeutic target for atrial fibrillation and cardioembolic stroke, and further research is warranted to decipher the mechanism through which it protects against cardiovascular and neurological diseases.

In addition, we have identified CD6, a lymphocyte surface receptor, associated with an increased risk of any stroke. CD6 is a pan T-cell marker[47,48], and is involved in T-cell proliferation and activation through its interaction with ALCAM (activated leucocyte cell adhesion molecule)[49]. The interaction of CD6 and ALCAM is required to promote an inflammatory T-cell response[50]. Interestingly, ref. 51 found that acute ischaemic stroke patients with upregulated ALCAM at admission had a significantly poorer survival rate ($P < 0.001$). Given this interaction and that the recruitment of leucocytes and platelets is widely

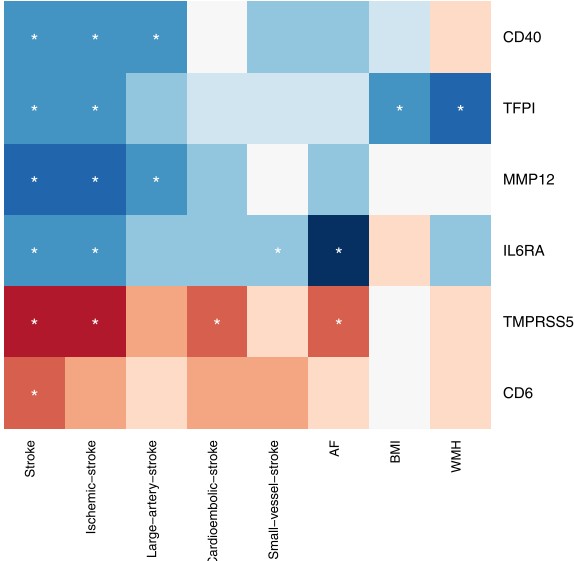

**Fig. 5 | Effect sizes (*Z*-score) of six potential causal proteins on stroke outcomes and causal risk factors for stroke.** MR analyses of the effect of proteins on stroke and stroke risk factors. Colours in each lattice of the heatmap represent the effect size (*Z*-score), with genetically predicted increased protein level associated with a higher risk of outcomes coloured in brown and lower risk of outcomes coloured in blue. The darker the colour the larger the effect size. *Indicates that the causal association is significant, which passed Bonferroni correction of $P_{causalEstimate\_IVW}$ $\leq 0.05/308 = 1.61 \times 10^{-4}$ and passed sensitivity tests with $P_{Qstat} \geq 0.05$ and $P_{EggerIntercept} \geq 0.05$.

regarded as a pivotal step in the inflammatory response associated with cerebral ischaemia[52,53], together with our finding that CD6 is associated with stroke, further investigation of CD6 in the context of stroke is justified.

When comparing MR results using pQTLs derived from two partly complementary techniques [the Olink (antibody-based assay, current study) and the SomaScan (aptamer-based assay, measured in ~3000 participants from the INTERVAL study[12])], we found that 257 proteins (out of 357) were measured by both platforms. Of which, 99% (255 out of 257) were found to have consistent results on stroke outcomes in both platforms and TFPI and TMPRSS5 were unique to Olink (Supplementary Data 16). Genetic variants in the *IL6R* region are associated with the risk of inflammatory-related diseases[25], coronary heart disease[54], stroke[55], atrial fibrillation[56] and rheumatoid arthritis[57]. Moreover, IL6R is the target of an FDA-approved therapy (Tocilizumab) for the treatment of several diseases, e.g. rheumatoid arthritis and systemic juvenile idiopathic arthritis. Phase II clinical trials testing tocilizumab for the therapy of non-ST elevation myocardial infarction have reported promising results[58] and a phase III clinical trial testing Ziltivekimab in cardiovascular disease and chronic kidney disease has recently started (NCT05021835).

To avoid violating the MR assumptions, we performed various sensitivity analyses. We used LD clumping at $R^2 \leq 0.1$ for pQTLs with $P \leq 5.0 \times 10^{-8}$ to choose instruments for each plasma protein level. However, concern[59] has been raised about the independency of the variants used as instrumental variables leading to violation of the InSIDE (instrument strength independent of direct effect) assumption of the MR-Egger method used. Therefore, we performed several sensitivity analyses to validate the robustness of the instrumental variables used in the MR analysis. Firstly, we performed MR analyses adjusting for the correlation of the variants used and obtained consistent and similar causal effect estimates to those obtained without adjusting for the correlation (Supplementary Data 4). Secondly, we performed conditional analysis and fine-mapping analysis to obtain instrumental variables for the six potential causal proteins, and we obtained consistent MR results (Supplementary Data 6 and Supplementary Fig. 7). Finally, colocalization analyses across the genetic associations with protein levels and stroke outcomes showed they were likely to have shared causal variants across these traits, supporting the validity of instrumental variables and the causal protein associations (Supplementary Data 8). We acknowledge, where there is evidence that the pQTL and the genetic variants associated with the outcome are shared, strengthens the support for the MR findings. However, we recognise that lack of colocalization evidence does not invalidate the findings as there is a high false negative rate with colocalization methodologies (typically around 60%)[60].

The Olink assay[61] used in our study measures the bulk concentration of protein in plasma. However, because this assay cannot distinguish free from bound protein or active from inactive, only limited mechanistic insights can be made. Due to the limited capture of the human proteome (1.5% of all known proteins), we could not evaluate the effects of all proteins within the same family or all proteins encoded within the same genomic region. For instance, we found that TMPRSS5 was a potential novel drug target for cardioembolic stroke, while other proteins in the Type II transmembrane serine protease family (TTSPs) that play crucial roles in cardiac functions[43,62] could not be evaluated. Thus, a targeted study of the TTSP family is warranted to comprehensively evaluate their effects on cardiovascular and neurological traits. We acknowledge there are limitations regarding the current multiplex platform for protein abundance measurements. For instance, IVs linking to a protein-altering variant (PAV) can influence the measurement of the protein binding affinity ('abundance'), leading to disconnection of the protein abundance and the function of the mutant protein[63]. But *cis*-eQTL (**e**xpression **Q**uantitative **T**rait **L**oci) are less likely to influence the protein abundances as measured by antibody-based Olink assays compared to the Somalogic aptamer-based assays[63]. A recent proteome study[15] using Somalogic aptamer-based platform showed that *cis*-eQTL were less likely to affect the protein binding. Indeed, if a significant pQTLs was in high LD with *cis*-eQTL, it was less likely to lead to binding artefacts. To test whether the IV(s) for the six target proteins were *cis*-eQTL for their protein-encoding gene, we sought eQTL from GTEx V8 and identified that IV(s) for five of the proteins (not MMP12) were significant eQTL in at least one tissue. The effect direction of these variants on gene expression and protein levels were consistent for TFPI, CD40, CD6 and TMPRSS5, except for IL6R. The diverse effect direction of variant on *IL6R* gene expression and IL6RA protein level is likely due to the measurement of IL6RA. The Olink assay only captures the circulating free IL6RA while all isoforms of *IL6R* transcripts are captured collectively by the gene expression measurements in tissues (blood and artery).

Our results highlight potential targets of future therapies for stroke outcomes and illustrates the relevance of proteomics in identifying drug targets. Further research is necessary to assess the viability of the six identified proteins as drug targets for stroke treatment. Additional drug targets may be uncovered as more comprehensive proteomics platforms become available and more diverse non-European ancestry populations are increasingly studied. Finally, there is an increasing need for similarly comprehensive proteomics across different tissues and organs to evaluate tissue- or organ-specific protein effects.

## Methods

The overall study design is illustrated in Fig. 1. Details of the methods and study participants are provided below.

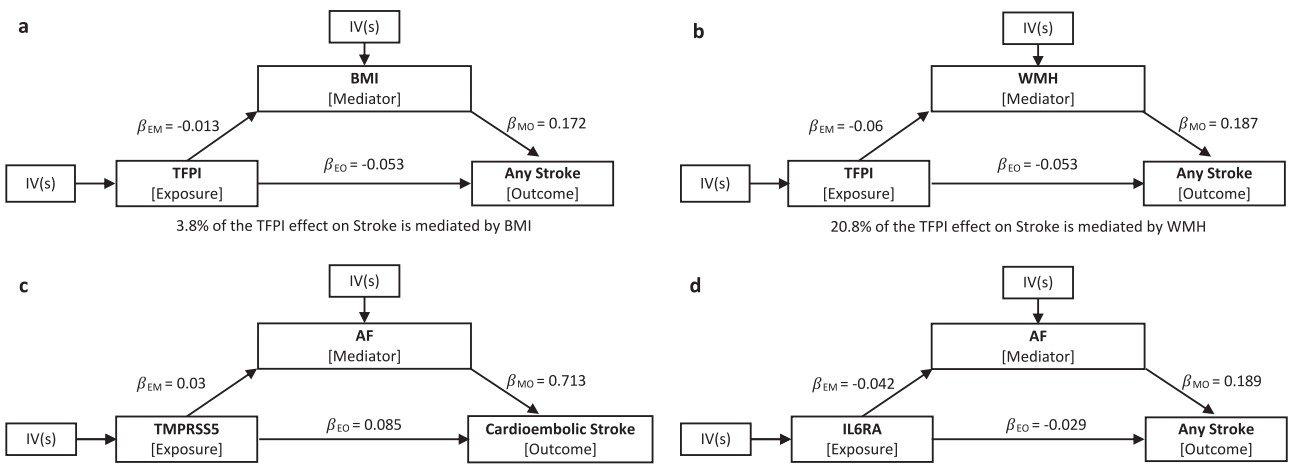

**Fig. 6 | Mediation effects of protein on stroke via risk factors.** Mediation analyses to quantify the effects of three proteins on stroke outcomes via risk factors. **a** TFPI effect on stroke mediated by BMI. **b** TFPI effect on stroke mediated by WMH; **c** TMPRSS5 effect on cardioembolic stroke mediated by AF; **d** IL6RA effect on stroke mediated by AF. $\beta_{EM}$ effects of exposure on mediator, $\beta_{MO}$ effects of mediator on outcome, $\beta_{EO}$ effects of exposure on outcome. BMI body mass index, WMH white matter hyperintensity, AF atrial fibrillation.

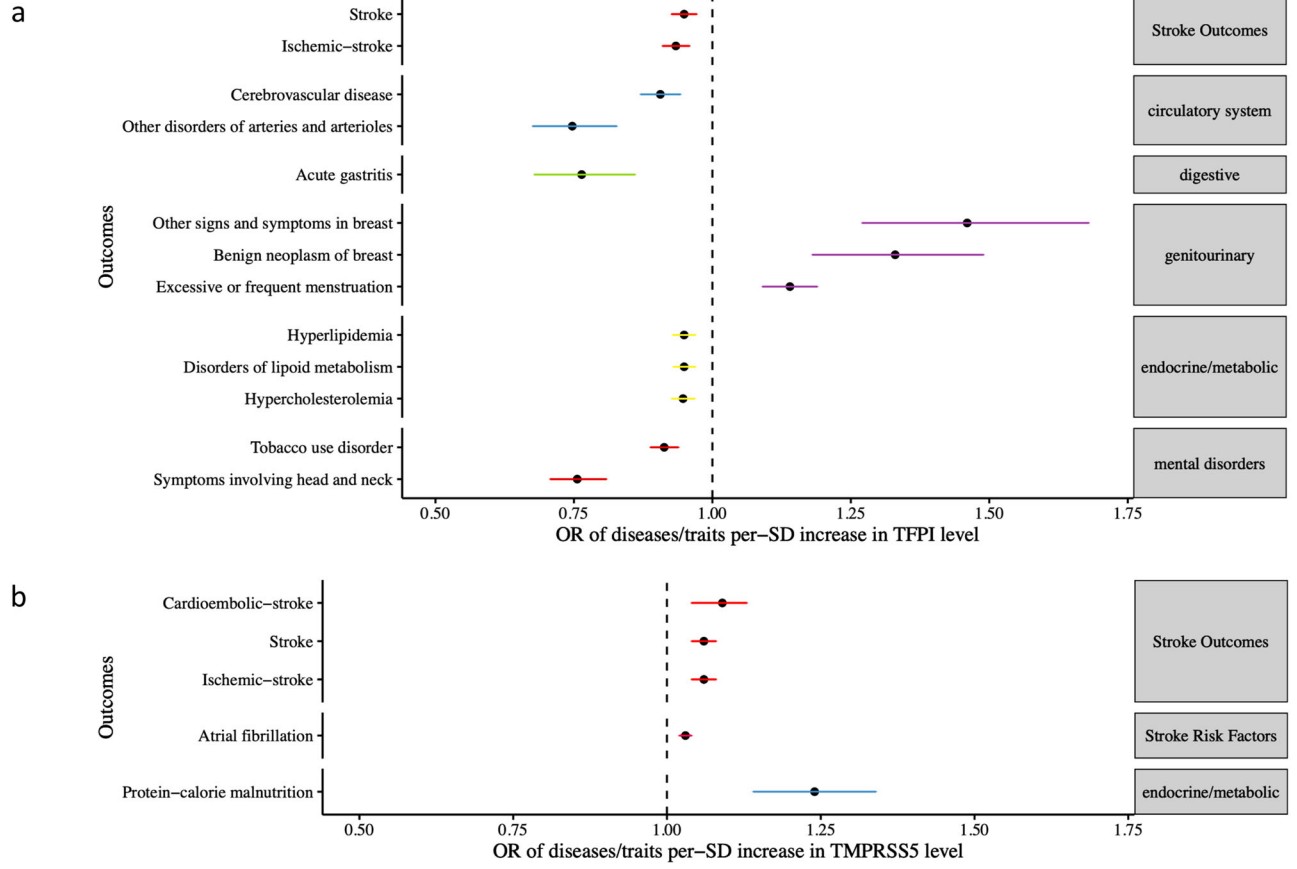

**Fig. 7 | Potential on-target effects of stroke-associated proteins.** Forest plots illustrating the potential on-target effects associated with causal proteins revealed by Phe-MR analysis for TFPI (**a**) and TMPRSS5 (**b**). In general, results can be perceived as the effects of per-SD higher circulating protein level on each phenotype. If the effect direction of the target protein on the phenotype is consistent with that on stroke outcomes, it represents 'beneficial' additional indications through the intervention of circulating protein level. Conversely, opposing effect directions of the target protein on the phenotype and stroke represents 'deleterious' side-effects. For example, a higher level of TFPI is associated with a lower risk of ischaemic stroke and so phenotypes with OR <1 represents 'beneficial effects', OR >1 represents 'deleterious effects' when the hypothetical intervention increases TFPI levels. Only significant associations that passed Bonferroni correction ($P \leq 0.05/6/784 = 1.06 \times 10^{-5}$) were plotted. See Supplementary Data 14 for more clinical information on the ICD code phenotypes. The dots are the causal estimates on the OR scale, and the whiskers represent the 95% confidence intervals for these ORs.

## Proteomic profiling and quality control

A subset of 4994 blood donors at a mean age of 61 years (SD 6.7 years) enroled in the INTERVAL BioResource[21], were processed for proteomic profiling using the Olink Proseek® Multiplex platform by four high-throughput, multiplex immunoassays: Inflammatory I (INF1), Cardiovascular II (CVD2), Cardiovascular III (CVD3) and Neurology I (NEURO) (Olink Bioscience, Uppsala, Sweden). Each panel enables the simultaneous measurement of 92 proteins through relative quantification using the proximity extension assay (PEA) Technology[61], in which each pair of oligonucleotide-labelled antibodies ('probes') are allowed to bind to their respective target present in the sample and trigger extension by DNA polymerase. DNA barcodes unique to each protein are then amplified and quantified using a standard real-time polymerase chain reaction (PCR). Default pre-processing of the proteomic data by Olink included applying median centring normalisation between plates, where the median is centred to the overall median for all plates, followed by log2 transformation to provide normalised protein expression (NPX) values. Further details on the Olink proteomic data processing can be found at http://www.olink.com. Probes were labelled using Uniprot identifiers, which we re-mapped to the HUGO gene name nomenclature for the (cis-) gene encoding the relevant protein. All protein names and descriptions are provided in Supplementary Data 1.

Samples that failed standard Olink quality control metrics were removed. 4902, 4947, 4987 and 4660 samples passed quality control for the 'INF1', 'CVD2', 'CVD3' and 'NEURO' panels, respectively. According to the manufacturer's recommendation, we also removed four proteins (HAGH, BDNF, GDNF and CSF3) in the 'NEURO' panel and one protein (GDNF) in the 'INF1' panel due to high levels of missingness.

## Proteome GWAS

The INTERVAL study[21] was genotyped using the UK Biobank Affymetrix Axiom array (http://www.ukbiobank.ac.uk/scientists-3/uk-biobank-axiom-array/), and imputed to 1000 Genomes Phase 3-UK10K combined reference panel, employing the PBWT imputation algorithm[64]. Genetic data for the ~5000 participants with Olink proteomic profiling were extracted to test for the association of the genetic variants with plasma proteins. More details regarding the INTERVAL genetic data QC can be found here[65]. Within the ~5000 participant subset, we removed six related individuals (those individuals with pairwise values of twice the kinship coefficient (PI_HAT) > 0.1875 (removing the individuals with the lowest call rate from each pair). The final imputed dataset was additionally filtered for imputation quality (only retaining variants with an info score >0.4) and Hardy–Weinberg equilibrium (retaining variants with $P_{HWE} > 1 \times 10^{-4}$).

About 354 proteins (of 363) that passed quality control were taken forward for the GWAS. Normalised protein levels ('NPX') were regressed on sex, age, plate, time from blood draw to processing (in days), season (as a categorical variable: 'Spring', 'Summer', 'Autumn', 'Winter') and batch when appropriate. The season is used as a covariate as it is a determinant of the levels of some proteins[66], so removing this non-genetic source of variation will improve the power to detect genetic association signals. The residuals were then rank-inverse normalised. Linear regression of the rank-inversed normalise residuals on genotype was carried out in SNPTEST v.2.5.2[67], with the first three components of multi-dimensional scaling as covariates to adjust for ancestry. Only proteins with at least one SNP with an association $P$ value passing the genome-wide significant threshold ($P \le 5.0 \times 10^{-8}$) were kept, resulting in 308 proteins for MR analyses.

## Genetic variants associated with proteins

For each plasma protein, cis- and trans- pQTLs from its corresponding GWAS were used as genetic instruments. We followed

these steps to select pQTLs instruments: (i) we obtained SNPs that were also tested in the MEGASTROKE GWAS of stroke outcomes (see below), (ii) we performed linkage disequilibrium (LD) clumping using PLINK 1.90 (www.cog-genomics.org/plink/1.9/)[68] to obtain approximately independent SNPs for each protein. In brief, the LD clumping algorithm groups SNPs in LD ($r^2 \ge 0.1$ in 4994 European ancestry participants from the INTERVAL study[21,65]) within ±1 MB of an index SNP (SNPs with association $P \le 5 \times 10^{-8}$). Analyses assessing sensitivity to the $r^2 \ge 0.1$ LD threshold are detailed below. The algorithm loops through all index SNPs, beginning with the smallest $P$ value and only allowing each SNP to appear in one clump. The final output, therefore, contains the most significant protein-associated SNPs for each LD-based clump across the genome. We split pQTLs variants into cis-pQTLs (±1 MB window of the gene encoding the target protein) and trans-pQTLs (outside the ±1 MB window). We then performed MR in a two-step approach. Our primary analysis was restricted to cis-pQTLs. Having performed MR restricted to cis-pQTLs only as IVs, we broadened the analysis to consider the effects of adding in trans-pQTLs as IVs. We estimated the variance of each protein explained by its IVs by calculating the $R^2$[69] and the strength of each IV by the F-statistic[70]. Summary association statistics of all the instrumental variables (IVs) for the 15 stroke-associated proteins are provided in Supplementary Data 2.

To assess the robustness of the $r^2 \ge 0.1$ thresholds for IV selection, we performed two additional sensitivity analyses (Supplementary Data 5) for proteins of interest to verify the robustness of the MR causal relationship: (1) we performed conditional analysis to derive conditionally independent variants as IVs using the FINEMAP software package[71] with–cond flag; (2) we performed fine-mapping to obtain variants in the 95% credible set as IVs using FINEMAP software package[71] with ---sss flag.

## Genetic variants associated with stroke and its risk factors

The primary outcomes were the risk of stroke and its subtypes. Genetic association estimates for stroke outcomes were obtained from the MEGASTROKE consortium, a large-scale international collaboration launched by the International Stroke Genetics Consortium (ISGC). A detailed description of the study design and characteristics of study participants were provided in the original publication[22]. To reduce confounding by population stratification, we extracted estimates for the associations of the protein IVs with stroke and its subtypes restricted to individuals of European ancestry (40,585 cases and 406,111 controls). The primary outcomes for this study were any stroke (including both ischaemic and haemorrhagic stroke; AS, $N_{cases}$ = 40,585), any ischaemic stroke (IS, $N_{cases}$ = 34,217) and the three aetiologic ischaemic stroke subtypes: large artery stroke (LAS, $N_{cases}$ = 4373), cardioembolic stroke (CES, $N_{cases}$ = 7193) and small vessel stroke (SVS, $N_{cases}$ = 5386). Summary-level data (beta coefficients and standard errors) for the associations of the five stroke outcomes were obtained from the MEGASTROKE GWAS http://www.megastroke.org/index.html.

The secondary outcomes we considered were stroke risk factors, which were selected from a literature review[1,72–74]. We performed a Pubmed search using the search terms, stroke, ischaemic stroke, haemorrhage stroke, risk factors, that identified 1494 manuscripts. These were reduced to 90 by applying filters of: Full text, Guideline, Meta-Analysis, Review and Systematic Review, in the last 10 years. We also referred to the following stroke website: https://www.stroke.org.uk/. We sought well-powered publicly available GWAS summary statistics for these risk factors and removed risk factors that did not have GWAS data, such as air pollution. The remaining seven risk factors were considered for the two-sample MR analyses, including blood pressure (BP)[75], atrial fibrillation (AF)[76], type 2 diabetes (T2D)[77], white matter hyperintensity (WMH)[78], body mass index (BMI)[79], alcohol consumption and

smoking behaviour[80]. We used the same pQTLs as IVs for the secondary outcomes as for the primary outcomes. The SNP-outcome effects for all the above risk factors were obtained from previously published GWASs when available. Table 1 provides full details of the data sources and sample size for these GWASs.

## Systematic MR screening for causal proteins of stroke and stroke risk factors

We used two-sample MR[81–83] to estimate the associations between genetically predicted protein levels and target outcomes (stroke, stroke risk factors and potential adverse effects or additional indications). Two-sample MR[84] is where the genetic associations with the risk factor are derived in one cohort (e.g. pQTLs from INTERVAL) and the association of these genetic variants with the outcome is tested in a second cohort (e.g. stroke GWAS from MEGASTROKE). Two-sample MR allows the evaluation of causal effects using summary genetic association data, negating the need for individual participant data. The MR approach was based on the following assumptions: (i) the genetic variants used as an instrumental variable (IV) are associated with target exposure, i.e. protein levels; (ii) there are no unmeasured confounders of the associations between genetic variants and outcome; (iii) the genetic variants are associated with the outcome only through changes in the exposure, i.e. no pleiotropy.

After extracting the association estimates between the variants and the exposures or the outcomes, we harmonised the direction of estimates by effect alleles, and applied Wald's ratio method to estimate the causal effects when there was only one IV available for target exposure. If more than one IV was available, we applied the inverse-variance weighting (IVW) method, either in a fixed-effect framework (IVs ≤3) or in a multiplicative random-effect meta-analysis framework (IVs >3)[81]. We chose 3 as a cut-off for the random-effects model, as with >3 variants, there is potential for some heterogeneity within instrumental variables. (The multiplicative random-effects model allows for heterogeneity between causal estimates targeted by the genetic variants by allowing over-dispersion of the regression model.) Additionally, MR-Egger was applied for causal estimation when there were ≥3 IVs available[85]. We also performed several sensitivity analyses to assess the robustness of our results to potential violations of the MR assumptions, given these analyses have different assumptions for validity: (i) heterogeneity was estimated by the Cochran Q test;[81] (ii) horizontal pleiotropy was estimated using MR-Egger's intercept;[85] (iii) influential outlier IVs due to pleiotropy were identified using MR Pleiotropy Residual Sum and Outlier (MR-PRESSO)[86]; (iv) reverse MR was used to eliminate spurious results due to reverse causation. Additionally, the contamination mixture method[87], which can explicitly model multiple potential causal estimates and therefore infer multiple causal mechanisms associated with the same risk factor that affects the outcome to different degrees, was also used to calculate the MR estimates. Although these methods may have different assumptions and statistical power, the rationale for using them is that if they give a similar conclusion, this provides greater certainty in inferring that any positive results are unlikely to be driven by violation of the MR assumptions.

We employed a two-sample MR framework incorporating the sensitivity analyses for both primary MR (proteins → stroke outcomes), two-step MR (Step-1 MR: stroke risk factors → stroke outcomes; Step-2 MR: proteins → stroke risk factors) and Phe-MR (proteins → PheWAS). The MR methods applied in each of the MR settings depend on the number of IVs for each exposure. Effects on binary outcomes (i.e. stroke, AF, T2D, smoking initiation/cessation) are reported as odds ratios (ORs) with their 95% confidence intervals (CIs) scaled to a one standard deviation (SD) higher plasma protein level. Effects on quantitative outcomes (i.e. BP, WMH, BMI) are reported as the effect size (95% CI) scaled to a 1-SD higher plasma protein levels. All statistical tests were two-sided and

considered statistically significant at $P_{\text{CausalEstimate}} \leq 1.62 \times 10^{-4}$ (Bonferroni-adjusted for 308 proteins: $0.05/308 = 1.62 \times 10^{-4}$), $P_{\text{Q-stat}} \geq 0.05$, $P_{\text{Egger-Intercept}} \geq 0.05$ and $P_{\text{GlobalTest}} \geq 0.05$. The MR analyses were conducted using *MendelianRandomization* (Version: 0.4.2)[82], *TwoSampleMR* (Version: 0.4.22)[83] and *MR-PRESSO* (Version: 1.0)[86] packages in R 3.5.1 (R Foundation, www.R-project.org). Plots were generated using various R packages including *ggplot2* (Version: 3.2.0), *forestplot* (Version: 1.9) and *PheWAS* (Version: 0.99.5-4).

## Multi-trait colocalization analyses

As the instruments used in the current setting were identified based on their statistical associations with the protein level, we conducted another sensitivity analysis–colocalization, to investigate whether the genetic associations with both protein and phenotypes shared the same causal variants. We conducted colocalization analysis for each of the six proteins associated with one or more of the stroke outcomes to investigate whether the protein level and stroke outcome genetic associations are due to the same causal variants. We estimated the posterior probability (PP) of multiple traits sharing the same causal SNP simultaneously using a multi-trait colocalization (HyPrColoc) method[88]. We also applied HyPrColoc for three proteins that showed causal relationships with stroke risk factors across multiple traits, i.e. for TFPI, HyPrcoloc was applied to TFPI pQTLs and GWAS of stroke, ischaemic stroke, WMH and BMI; for TMPRSS5, HyPrColoc was applied to TMPRSS5 pQTLs and GWAS of stroke, ischaemic stroke, cardioembolic stroke and AF; for IL6R, HyPrColoc was applied to IL6R pQTLs and GWAS of stroke, ischaemic stroke, cardioembolic stroke and AF. Furthermore, we extended HyPrColoc analyses to 39 proteins that were associated with at least one stroke risk factor(s). HyPrColoc extends the established coloc methodology[89] by approximating the true posterior probability of colocalization with the posterior probability of colocalization at a single causal variant and a small number of related hypotheses. If all traits do not share a causal variant, HyPrColoc employs a novel branch-and-bound selection algorithm to identify subsets of traits that colocalize at distinct causal variants at the locus. We used uniform priors for the primary analysis. We also performed a sensitivity analysis with non-uniform priors to access the choice of priors, which used a conservative trait-level prior structure with $P = 1 \times 10^{-4}$ (prior probability of an SNP being associated with one trait) and $\gamma = 0.98$ (1-prior probability of an SNP being associated with an additional trait given that the SNP is associated with at least one other trait), i.e., 1 in 500,000 variants are expected to be causal for two traits.

Variants within a ±1 Mb window around the pQTLs with the smallest $P$ value, with imputation (INFO)-score ≥0.8 and minor allele frequency (MAF) ≥0.01, were included. All variants across each of the datasets were harmonised to the same effect alleles prior to colocalization analyses. We conducted the colocalization analysis using the 'HyPrColoc' R package[88].

## Mediation analysis

For proteins that causally associate with both stroke and risk factors, we conducted a mediation analysis to quantify the effects of proteins on stroke outcomes via risk factors. The 'total' effect of exposure on outcome includes both 'direct' effect and any 'indirect' effect via one or more mediators. In this study, the total effect is captured by a standard univariable MR analysis–the primary MR. To decompose direct and indirect effects, we used results from two-step MR and chose the Product method to estimate the beta of indirect effect and the Delta method to estimate the standard error (SE) and confidence interval (CI)[90].

## Phenome-wide MR (Phe-MR) analysis of 784 phenotypes for target proteins

We expanded the exploration of side-effects for the six stroke-associated proteins to include non-stroke phenotypes by performing

Phe-MR analyses for a range of diseases. We used summary statistics for SNP-outcome effects calculated using the UK Biobank cohort ($N \le 408,961$) by ref. 91, who performed GWAS using the Scalable and Accurate Implementation of GEneralized mixed model (SAIGE v.0.29) method[91] to account for unbalanced case-control ratios. They defined disease outcomes based on 'PheCodes', a system developed to organise International Classification of Diseases and Related Health Problems (ICD-9/−10) codes into phenotypic outcomes suitable for systematic genetic analysis of numerous disease traits[23,91]. Outcomes with fewer than 500 cases were excluded due to statistical power, leaving 784 diseases for Phe-MR analyses (Supplementary Data 14). SNP-outcome associations were downloaded from SAIGE GWAS[91] (https://www.leelabsg.org/resources). pQTLs were derived from the same proteome GWAS as in the primary analysis with stroke subtypes. Phe-MR findings can be interpreted as the risk/protective effect per-SD increase in the plasma protein level, same as with primary stroke outcomes. That is, if the effect direction of the additional indication is consistent with the effect direction in Stroke, the identified protein that is therapeutically targeted for the treatment of stroke may also be 'beneficial' for the additional indication, and vice versa. MR causal effects are considered statistically significant at $P \le 1.06 \times 10^{-5}$ (Bonferroni-adjusted for six proteins and 784 phenotypes: 0.05/6/784).

### Reporting summary

Further information on research design is available in the Nature Research Reporting Summary linked to this article.

## Data availability

Part of the INTERVAL Olink proteome GWAS summary statistics have already been published as part of larger collaborative meta-analyses from the SCALLOP Consortium (Folkersen 2020; https://doi.org/10.5281/zenodo.2615265) and the others are the subject of forthcoming GWAS discovery manuscripts and will be made available upon publication. Data were available upon request from the corresponding author. URLs for GWAS summary statistics used for Mendelian randomisation and colocalization analyses are available as follows: stroke outcomes (Malik 2018; http://www.megastroke.org/index.html), blood pressure (Surendran 2020; https://app.box.com/s/1ev9iakptips70k8t4cm8j347if0ef2u), atrial fibrillation (Nielsen 2018; http://csg.sph.umich.edu/willer/public/afib2018), type 2 diabetes (Mahajan 2018; http://diagram-consortium.org/), white matter hyperintensity (Persyn 2020; http://cerebrovascularportal.org/informational/downloads), body mass index (Pulit 2019; https://doi.org/10.5281/zenodo.1251813), alcohol consumption and smoking behaviour (Liu 2019; https://genome.psych.umn.edu/index.php/GSCAN), UK Biobank SAIGE GWAS (Zhou 2018, https://www.leelabsg.org/resources). Table 1 provides further information on the genetic data resources. All other data that support the findings of this study are available in the Supplementary Data.

## Code availability

We used publicly available software (URLs are listed below). PBWT imputation algorithm, https://github.com/richarddurbin/pbwt; SNPTEST v.2.5.2, https://www.well.ox.ac.uk/~gav/snptest/; PLINK 1.90, www.cog-genomics.org/plink/1.9/; FINEMAP v1.4, http://www.christianbenner.com/; LDstore v2.0, http://www.christianbenner.com/; HyPrColoc v1.0.0, https://doi.org/10.5281/zenodo.4293559. Mendelian Randomisation with R packages 'TwoSampleMR' version 0.4.22, 'MendelianRandomization' version 0.4.1 and 'MR-PRESSO' version 1.0. We used R (version 3.5.1) extensively to analyse data and create plots. Code used to perform Mendelian randomisation analyses is available at https://doi.org/10.5281/zenodo.7042044.

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

## Acknowledgements

This work and L.C. were funded by a programme grant from the British Heart Foundation (RG/16/4/32218). P.S. was supported by a Rutherford Fund Fellowship from the Medical Research Council (MR/S003746/1). B.P. and S.K. are funded by a British Heart Foundation Programme grant (RG/18/13/33946). E.P. was funded by the EU/EFPIA Innovative Medicines Initiative Joint Undertaking BigData@Heart grant 116074 and the NIHR BTRU in Donor Health and Genomics (NIHR BTRU-2014-10024) and is funded by the NIHR BTRU in Donor Health and Behaviour (NIHR203337). E.Y.-D. was funded by the Isaac Newton Trust/Wellcome Trust ISSF/University of Cambridge Joint Research Grants Scheme. S.B. is supported by a Sir Henry Dale Fellowship jointly funded by the Wellcome Trust and the Royal Society (204623/Z/16/Z). J.M.M.H. was funded by a BHF Programme Grant (RG/13/13/30194) and the NIHR Cambridge Biomedical Research Centre (BRC-1215-20014) [1]. C.M.L. is funded by the NIHR Maudsley Biomedical Research Centre at South London and Maudsley NHS Foundation Trust and King's College London. H.S.M. is supported by an NIHR Senior Investigator award, and his work is supported by the Cambridge Universities NIHR Comprehensive Biomedical Research Centre. J.D. holds a British Heart Foundation Professorship and an NIHR Senior Investigator Award [1]. Participants in the INTERVAL randomised controlled trial were recruited with the active collaboration of NHS Blood and Transplant England (www.nhsbt.nhs.uk), which has supported fieldwork and other elements of the trial. DNA extraction and genotyping were co-funded by the National Institute for Health and Care Research (NIHR), the NIHR BioResource (http://bioresource.nihr.ac.uk) and the NIHR Cambridge Biomedical Research Centre (BRC-1215-20014) [1]. The Olink® Proteomics assays (Neurology panel) were funded by Biogen, Inc. (Cambridge, MA, USA). The academic coordinating centre for INTERVAL was supported by core funding from the: NIHR BTRU in Donor Health and Genomics (NIHR BTRU-2014-10024), NIHR BTRU in Donor Health and Behaviour (NIHR203337), UK Medical Research Council (MR/L003120/1), British Heart Foundation (SP/09/002; RG/13/13/30194; RG/18/13/33946) and NIHR Cambridge BRC (BRC-1215-20014) [1] and funding from the EC-Innovative Medicines Initiative (BigData@Heart). A complete list of the investigators and contributors to the INTERVAL trial is provided in reference [2]. The academic coordinating centre would like to thank blood donor centre staff and blood donors for participating in the INTERVAL trial. This work was supported by Health Data Research UK, which is funded by the UK Medical Research Council, Engineering and Physical Sciences Research Council, Economic and Social Research Council, Department of Health and Social Care (England), Chief Scientist Office of the Scottish Government Health and Social Care Directorates, Health and Social Care Research and Development Division (Welsh Government), Public Health Agency (Northern Ireland), British Heart Foundation and Wellcome. This research has been conducted using the UK Biobank Resource under Application Number '20480'. [1] The views expressed are those of the author(s) and not necessarily those of the NIHR, NHSBT or the Department of Health and Social Care. [2] Di Angelantonio E, Thompson SG, Kaptoge SK, Moore C, Walker M, Armitage J, Ouwehand WH, Roberts DJ, Danesh J, INTERVAL Trial Group. Efficiency and safety of varying the

frequency of whole blood donation (INTERVAL): a randomised trial of 45,000 donors. Lancet. 2017 Nov 25;390(10110):2360-2371.

## Author contributions

L.C. performed the main analyses and wrote the initial draft of the manuscript. J.E.P. and B.P. performed proteome GWAS analyses and contributed to data interpretation. M.T., P.S., S.K. and E.Y.-D. provided outcome GWAS data and analytical support. S.B. advised on Mendelian randomisation analyses. E.D.A, D.J.R., N.A.W., W.H.O., and J.D. lead the INTERVAL BioResource. E.P., H.S.M., and C.M.L. provided WMH GWAS. H.S.M. provided the clinical perspective on stroke research. P.G.B. critically reviewed and revised the manuscript. A.S.B. supervised the proteomics study and revised the manuscript. J.M.M.H. designed and supervised the project and drafted the manuscript. All authors critically reviewed and approved the final version of this manuscript.

## Competing interests

J.M.M.H., L.C., M.T. and E.Y.-D. became full-time employees of Novo Nordisk Ltd. during the drafting of this manuscript. J.D. serves on scientific advisory boards for AstraZeneca, Novartis and UK Biobank, and has received multiple grants from academic, charitable and industry sources for the submitted work. A.S.B. reports institutional grants from AstraZeneca, Bayer, Biogen, BioMarin, Bioverativ, Novartis, Regeneron and Sanofi and personal fees from Novartis. P.G.B. is a full-time employee of Biogen Inc. The remaining authors declare no competing interests.

## Additional information

[1]British Heart Foundation Cardiovascular Epidemiology Unit, Department of Public Health and Primary Care, University of Cambridge, Cambridge, UK. [2]Department of Genetics, Novo Nordisk Research Centre Oxford, Oxford, UK. [3]Department of Immunology and Inflammation, Faculty of Medicine, Imperial College London, London, UK. [4]Department of Medical and Molecular Genetics, King's College London, London, UK. [5]Cambridge Baker Systems Genomics Initiative, Department of Public Health and Primary Care, University of Cambridge, Cambridge, UK. [6]Clinical Pharmacology, William Harvey Research Institute, Queen Mary University of London, London, UK. [7]Rutherford Fund Fellow, Department of Public Health and Primary Care, University of Cambridge, CB1 8RN Cambridge, UK. [8]British Heart Foundation Centre of Research Excellence, University of Cambridge, Cambridge, UK. [9]National Institute for Health and Care Research Blood and Transplant Research Unit in Donor Health and Behaviour, University of Cambridge, Cambridge, UK. [10]Health Data Research UK Cambridge, Wellcome Genome Campus and University of Cambridge, Cambridge, UK. [11]Health Data Science Research Centre, Human Technopole, Milan, Italy. [12]NHS Blood and Transplant-Oxford Centre, Level 2, John Radcliffe Hospital, Oxford, UK. [13]Radcliffe Department of Medicine, University of Oxford, John Radcliffe Hospital, Oxford, UK. [14]NHS Blood and Transplant, Cambridge Biomedical Campus, Long Road, Cambridge, UK. [15]Department of Haematology, University of Cambridge, Cambridge, UK. [16]Wellcome Sanger Institute, Hinxton, UK. [17]Department of Human Genetics, Wellcome Sanger Institute, Hinxton, UK. [18]Social, Genetic and Developmental Psychiatry Centre, King's College London, London, UK. [19]R&D Translational Biology, Biogen, Inc., Cambridge, MA, USA. [20]Department of Clinical Neurosciences, University of Cambridge, Cambridge, UK. [21]Medical Research Council Biostatistics Unit, Cambridge Institute of Public Health, University of Cambridge, Cambridge, UK. ✉e-mail: jmmh2@medschl.cam.ac.uk

