## [Peer Review File · Nature Communications]

Systematic Mendelian randomization using the human plasma proteome to discover potential therapeutic targets for strokeREVIEWER COMMENTS

Reviewer #1 (Remarks to the Author):

Chen et al presented a very nice and comprehensive GWAS + MR study of plasma proteins on stroke (and stroke risk factors). The design is comprehensive, the GWAS data are novel, methods are state-of-the-art and robust. The interpretation of the results are quite fair.

In general, this manuscript make a very nice demonstration of how to conduct a nice MR study. However, the presentation of the work is not direct clear and can be improved. Some more additional analyses could be considered.

Major comments:

1. presentation of the study

To me, this study is a classic two-step / mediation MR study design (plus MR-PheWAS of top findings). Proteins as exposures, stroke risk factors as mediators and stroke (stroke sub-types) as outcomes. However, the current way of present this work makes me feel confused from time to time. For example, the second step, risk factor on stroke was not mentioned in the methods and only been presented in results. In addition, by reading over the whole manuscript a few times, I am not sure when coloc was applied. It is also not clear whether the GWAS of proteome was conducted in this study or somewhere else. If the GWAS was conducted in this study, better to present some top findings in the results (I believe the N.C. readers can easily understand GWAS results)? In general, the presentation of methods and results (e.g. sub-titles and orders of results) as well as Figure 1 can be refined so these section can guide readers through your nice story.

In a similar note, a figure summarise the top findings can be useful. e.g. a DAG shows how many risk factors showed potential mediation effect of the proteins-stroke associations.

In addition, the current abstract is in a MR style, which is fine. However, few key information was missing. (1) if a novel proteome GWAS in OLINK was conducted but not been mentioned in the abstract; (2) colocalization is important but not been mentioned.

2. results comparison with existing MR evidence.

A general question is how OLINK pQTL MR results on stroke compared with existing SOMALogic pQTL MR. This can be done quickly as a validation.

3. pleiotropy of pQTL instruments

Example such as ABO was mentioned in the results section. However, it is not clear how pleiotropic the instruments of the top six proteins are. e.g. how many proteins these instruments are associated with (e.g. check this in Sun et al or some recent dataset such as Zhang et al ARIC study/Finland/Decode datasets with better protein coverage)?

4. multiple testing

what are the genetic correlation among stroke subtypes? can these outcomes been treated as one signal variable when control for multiple testing?

5. selection of stroke risk factors

It is not clear how these risk factors have been selected. Is this from a literature review or search from pubmed database?

6. selection of top findings

currently, the top findings were selected based only on MR findings. Although a lot of MR methods had been applied (including some new ones such as the contamination mixture method). However, I am not sure how reliable these methods are when we only have a limited number of instruments (If these methods were only applied to risk factor against stroke, then please specify this in the methods.). To

me, 4 of 6 top protein-disease associations with coloc evidence should be considered as top findings.

7. Coloc

The HyPrColoc method was applied for proteins vs stroke sub-types, which is interesting. However, I feel a HyPrColoc of protein, stroke risk factors and stroke will be more meaningful. Especially for TFPI and TMPRSS5 effect on stroke.

In addition, IL6RA is an important protein isoform, why not apply Susie + coloc (or PWCoCo) for this case?

8. mediator (optional)

The current setting testing protein  risk factor  stroke. However, some recent studies showed that BMI causally influence protein levels (e.g. CRP). Will it be useful to also consider risk factor  protein  stroke. This may further increase the complexity of the study design. So just a suggestion.

Minor comments

Line 72, Put a reference if there is such a tissue specific eQTL study on IBD?

Line 74, some more proteome GWAS coming out recently in Science and Nature Genetics.

Line 80, a reference for MR vs RCT

Line 129, why season was controlled as a covariate?

Line 268, "causal associations" changes to "causal relationships"

Line 275 (and other places), not a big fan of define "significance" based on p value.

Line 296, as mentioned above, cis acting pQTLs could be pleiotropic too. need to do a test.

Line 363, very interesting finding, can do a mediation analysis to estimate the % of mediation effect.

Figure 3 & 4, put number of SNPs into the forest plots?

Figure 6, "A. Stroke" Is this a typo?

btw, please change the manuscript to a N.C. format, e.g. abstract and order of sections.

Reviewer #2 (Remarks to the Author):

Chen et al have investigated whether genetically determined blood levels of over 300 proteins are associated with ischemic stroke. Significant associations may be indicative of causal effects. They identify 6 proteins that are robustly associated to ischemic stroke. These are important results from a well-executed study, expanding our understanding of stroke etiology and providing new potential drug targets.

Major comments

Multiplex platforms such as O-link are not gold standard measure of protein levels. The authors do note that O-link cannot distinguish between free or bound protein nor active or inactive, but there may be differences beyond that just based on the technology. Could there be cis-eQTLs that influence binding of the O-link assay as opposed to protein levels? If so, would that create false positive MR associations or false negative MR associations? Could the authors discuss (in their limitations section) this topic in some more depth?

Suggest being more transparent about the number of instrumental variables that are behind the core 6 MR associations. If there are more than 3 instruments it would be nice to see the median-based, mode-based, and Egger-based MRs. Although MR-Egger was used according to the flowchart, I was not able to find references to it in the main text (except for references to the MR-Egger intercept).

Minor comments

"The association of TFPI, IL6RA and TMPRSS5 with stroke could be mediated by these risk factors, such as body mass index, white matter hyperintensity and atrial fibrillation." This sentence sounds a bit vague, as if you are stating a hypothesis instead of a result. Replace "could be" with something like "appeared to be"?

Does it make sense to use $5E-8$ as the significance threshold in the identification of pQTLs, when 354 proteins were tested?

The "phenome-wide MR" section in the methods focused on 6 stroke associated proteins currently comes before the section that describes the identification of these 6 proteins. Suggest moving it further back.

For the phenome-wide MR section of the results, you note that "If the effect direction of the protein on the disease or trait is the same as on stroke, the effect is considered "beneficial" and "deleterious" otherwise." For clarify I suggest using this beneficial/deleterious terminology in the next sentence (and throughout) as well.

Response to REVIEWER COMMENTS

Reviewer #1 (Remarks to the Author):

Chen et al presented a very nice and comprehensive GWAS + MR study of plasma proteins on stroke (and stroke risk factors). The design is comprehensive, the GWAS data are novel, methods are state-of-the-art and robust. The interpretation of the results are quite fair.

In general, this manuscript makes a very nice demonstration of how to conduct a nice MR study. However, the presentation of the work is not direct clear and can be improved. Some additional analyses could be considered.

We appreciate the reviewer's detailed comments and suggestions. We have updated the manuscript to clarify the story and add the details the reviewer requested. Furthermore, we have performed additional analyses as appropriate. We thank the reviewer for helping us to improve the clarity of the manuscript.

Major comments:

1. presentation of the study

To me, this study is a classic two-step / mediation MR study design (plus MR-PheWAS of top findings). Proteins as exposures, stroke risk factors as mediators and stroke (stroke sub-types) as outcomes. However, the current way of presenting this work makes me feel confused from time to time. For example,

1(a) the second step, risk factor on stroke was not mentioned in the methods and only been presented in results.

Response to 1(a): The second step describing the analysis of risk factors on stroke has been added to **Figure 1** (Provided below for your convenience) and the **Methods (Page 19, Line 444-453)**. Please also refer to reply to comment 5.

“ The secondary outcomes we considered were stroke risk factors, which were selected from a literature review ^{1,72-74}. We performed a Pubmed search using the search terms, stroke, ischemic stroke, haemorrhage stroke, risk factors, that identified 1494 manuscripts. These were reduced to 90 by applying filters of: Full text, Guideline, Meta-Analysis, Review, Systematic Review, in the last 10 years. We also referred to the following stroke website: <https://www.stroke.org.uk/>. We sought well-powered publicly available GWAS summary statistics for these risk factors and removed risk factors that did not have GWAS data, such as air pollution. The remaining 7 risk factors were considered for the two-sample MR analyses, including blood pressure (BP) ⁷⁵, atrial fibrillation (AF) ⁷⁶, type 2 diabetes (T2D) ⁷⁷, white matter hyperintensity (WMH) ⁷⁸, body mass index (BMI) ⁷⁹, alcohol consumption and smoking behaviour ⁸⁰. ”

1(b) In addition, by reading over the whole manuscript a few times, I am not sure when coloc was applied.

Response to 1(b): We have applied colocalization analyses (HyPrColoc) for six target proteins where we obtained evidence of an effect on stroke. Specifically, we have tested whether the genetic variants associated with circulating proteins levels are the same as the genetic variants associated with each of the five stroke outcomes, *i.e.* whether they share the same underlying causal variants. Colocalization analyses were also performed for the three proteins where there was evidence that the stroke risk factors mediated the protein effects on stroke outcomes. We have added the following to the **Methods (Page 21, line 510-518)**:

" We conducted colocalization analysis for each of the six proteins associated with one or more of the stroke outcomes to investigate whether the protein level and stroke outcome genetic associations are due to same causal variants. We estimated the posterior probability (PP) of multiple traits sharing the same causal SNP simultaneously using a multi-trait colocalization (HyPrColoc) method⁸⁸. We also applied HyPrColoc for three proteins showed causal relationships with stroke risk factors across multiple traits, *i.e.* for TFPI, HyPrColoc was applied to TFPI pQTLs and GWAS of stroke, ischemic stroke, WMH and BMI; for TMPRSS5, HyPrColoc was applied to TMPRSS5 pQTLs and GWAS of stroke, ischemic stroke, cardioembolic stroke, and AF; for IL6R, HyPrColoc was applied to IL6R pQTLs and GWAS of stroke, ischemic stroke, cardioembolic stroke and AF. "

1(c) It is also not clear whether the GWAS of proteome was conducted in this study or somewhere else. If the GWAS was conducted in this study, better to present some top findings in the results (I believe the N.C. readers can easily understand GWAS results)?

Response to 1(c): INTERVAL proteome GWAS of three Olink Panels, *i.e.* INF1, CVD3, and NEURO, are the subject of forthcoming GWAS discovery manuscripts so the GWAS discovery aspects will be explored more fully in those manuscripts than can be achieved in this MR-focused paper. The INTERVAL Olink GWAS of the CVD2 panel has already been published as part of a larger collaborative meta-analyses from the SCALLOP Consortium (<https://pubmed.ncbi.nlm.nih.gov/33067605/>). We include this manuscript in the references (Reference 17, line 622) and "Data Availability" section (Page 23, line 561-576).

1(d) In general, the presentation of methods and results (e.g. sub-titles and orders of results) as well as Figure 1 can be refined so these section can guide readers through your nice story.

Response to 1 (d): We have updated **Figure 1** to illustrate the study design and have now reflected the components of figure 1 in the sub-titles so the reader can more readily follow the flow. Headings for each result session are updated as below:

"

- Identification of stroke associated proteins
- Shared genetic associations with protein levels and risk of stroke
- Identification of likely casual stroke risk factors
- Identification of stroke risk factors associated proteins
- Mediation effect of proteins on stroke outcomes via risk factors
- Phenome-wide MR (Phe-MR) analysis of stroke-associated proteins

Fig. 1 Overview of this MR study.

Four O-link panels were used to measure plasma proteins in a subset of ~5000 samples from the INTERVAL study. Genetic variants associated with plasma protein levels were identified based on results from their corresponding GWAS. These genetic variants were then used as proxies for the protein level and tested their relationship with stroke outcomes (Primary MR), and with stroke risk factors that were associated with stroke outcomes (Secondary MR). Colocalization analyses were performed to test the shared genetic associations of protein level, stroke outcomes and risk factors. Mediation analyses by two-step MR were performed for proteins that potentially causally associated with both risk factors and stroke outcomes. We also tested the relationships of the potentially causal stroke proteins with 784 phenotypes in UK Biobank to test a broad spectrum of potential effects of hypothetical therapeutic agents for stroke. MR=Mendelian Randomization; IV=Instrumental Variable; AF=Atrial Fibrillation; BMI=Body Mass Index; WMH=White Matter Hyperintensity; T2D=Type 2 Diabetes; SBP=Systolic Blood Pressure; DBP=Diastolic Blood Pressure; PP=Pulse Pressure.

1(e) In a similar note, a figure to summarise the top findings can be useful. e.g. a DAG shows how many risk factors showed potential mediation effect of the proteins-stroke associations.

Response to 1(e): We have added a DAG plot to **Figure 6** as suggested (provided below for your convenience). We found three proteins where the risk factors partially mediated their effect on stroke outcomes.

Fig. 6 Mediation effects of protein on stroke via risk factors.

β_{EM} : effects of exposure on mediator; β_{MO} : effects of mediator on outcome; β_{EO} : effects of exposure on outcome. BMI=Body Mass Index; WMH=White Matter Hyperintensity; AF=Atrial Fibrillation.

1(f) In addition, the current abstract is in a MR style, which is fine. However, few key information was missing. (1) if a novel proteome GWAS in OLINK was conducted but not been mentioned in the abstract; (2) colocalization is important but not been mentioned.

Response to 1(f): We have reviewed the abstract and updated as the reviewer suggests:

“ We estimated the causal effects of 308 plasma proteins (using GWAS performed in 4,994 blood donors from the INTERVAL study) on stroke outcomes (MEGASTROKE GWAS) in a two-sample MR framework and assessed whether these associations could be mediated by stroke risk factors.

We found associations between genetically predicted plasma levels of TFPI, IL6RA, MMP12, CD40, TMPRSS5 and CD6, and stroke outcomes ($P \leq 1.62 \times 10^{-4}$). The genetic associations with stroke colocalized ($PP > 0.7$) with the genetic associations of four proteins (TFPI, TMPRSS5, CD6, CD40). ”

2. results comparison with existing MR evidence.

A general question is how OLINK pQTL MR results on stroke compared with existing SOMAlogic pQTL MR. This can be done quickly as a validation.

Thank you for the suggestion. We have now compared MR results using pQTLs derived from both platforms - Olink (current study) and SOMAlogic (measured in ~3,000 participants from the INTERVAL study (Sun et al 2018)) and found that 257 proteins (out of 357) were measured in both platforms. Of which, 99% of the proteins (255 out of 257) were found to have consistent MR results on stroke outcomes in both platforms, including two proteins (IL6RA and MMP12) that were significantly associated with stroke outcome(s). Moreover, we identified two proteins (TFPI &TMPRSS5) that showed significant association with stroke outcomes in Olink assays only. We have added these results to the **Discussions (Page 13, line 296-302)**.

“ When comparing MR results using pQTLs derived from two partly complementary techniques – the Olink (antibody-based assay, current study) and the SomaScan (aptamer-based assay, measured in ~3,000 participants from the INTERVAL study ¹²), we found that 257 proteins (out of 357) were measured by both platforms. Of which, 99% (255 out of 257) were found to have consistent results on stroke outcomes in both platforms and TFPI and TMPRSS5 were unique to Olink (**Supplementary Table 16**). ”

3. pleiotropy of pQTL instruments

Example such as ABO was mentioned in the results section. However, it is not clear how pleiotropic the instruments of the top six proteins are. e.g. how many proteins these instruments are associated with (e.g. check this in Sun et al or some recent dataset such as Zhang et al ARIC study/Finland/Decode datasets with better protein coverage)?

Thank you very much for the comments. We agree it is important to investigate potential pleiotropic effects of cis-pQTLs. We have investigated the associations of the IVs of our six target proteins with all other proteins that were measured using SomaScan in INTERVAL (Sun et al, 2018). We found IVs for both IL6RA and MMP12 were only associated with the protein level of IL6RA and MMP12, respectively. IVs for the remaining 4 proteins were not significantly associated with any proteins being measured in SOMAlogic (Sun et al, 2018). We therefore checked the recent DECODE data (Feringstad et al, 2021) for the other 4 proteins, and found 2 (CD6 and CD40) were associated with one additional protein respectively, but they are in low LD with the top signal of the other proteins and hence these are likely to be ‘shadow’ associations and not primary.

4. multiple testing

what are the genetic correlation among stroke subtypes? can these outcomes been treated as one single variable when control for multiple testing?

The MEGASTROKE paper did not report the genetic correlations among stroke subtypes, however 13 loci (out of 18 in total) were associated with both all stroke (AS) and ischemic stroke (IS), and the only one locus associated with small vessel stroke (SVS) was associated with both AS and IS. Although distinct associations with each of the stroke subtypes were reported, the samples for each of the stroke outcome GWAS were not independent, *i.e.* “All stroke” includes all samples from all other stroke outcomes.

Consequently, choosing a Bonferroni significance threshold would be too conservative. Consequently, we have now calculated the false discovery rate (FDR) corresponding to the detected MR causal associations. The FDR for the identified significant protein-stroke associations is ≤ 0.0046 , suggesting our results are robust and unlikely to be false. We have now added the FDR to the “MR: Proteins \rightarrow Stroke” for the stroke associated proteins to **Supplementary Table 3** in column “FDR”.

5. selection of stroke risk factors

It is not clear how these risk factors have been selected. Is this from a literature review or search from pubmed database?

The risk factors have been selected from a literature review. The following manuscripts and website were used to prioritise the listing of potential risk factors to investigate:

- Meschia, James F., et al. "Guidelines for the primary prevention of stroke: a statement for healthcare professionals from the American Heart Association/American Stroke Association." *Stroke* 45.12 (2014): 3754-3832.
- Feigin, Valery L., et al. "Global, regional, and national burden of neurological disorders, 1990–2016: a systematic analysis for the Global Burden of Disease Study 2016." *The Lancet Neurology* 18.5 (2019): 459-480.
- Feigin, Valery L., et al. "Global, regional, and national burden of stroke and its risk factors, 1990–2019: A systematic analysis for the Global Burden of Disease Study 2019." *The Lancet Neurology* 20.10 (2021): 795-820.
- <https://www.stroke.org.uk/>

We sought well-powered publicly available GWAS summary statistics data for these risk factors and removed risk factors that did not have GWAS data, such as air pollution. The remaining seven risk factors were considered for the two-sample MR analyses (blood pressure (BP), atrial fibrillation (AF), type 2 diabetes (T2D), white matter hyperintensity (WMH), body mass index (BMI), alcohol consumption and smoking behaviour). Furthermore, only risk factors that were found to be associated with stroke outcomes in our MR analyses were further explored with proteins. We have added selection of stroke risk factors to **Methods** as follows:

“ The secondary outcomes we considered were stroke risk factors, **which were selected from a literature review** ^{1,72-74}. We performed a Pubmed search using search terms, *i.e.* stroke, ischemic stroke, haemorrhage stroke, risk factors, that identified 1494 manuscripts. These were reduced to 90 by applying filters of: Full text, Guideline, Meta-Analysis, Review, Systematic Review, in the last 10 years. We also referred to the following stroke website: <https://www.stroke.org.uk/>. We sought well-powered publicly available GWAS summary statistics for these risk factors and removed risk factors that did not have GWAS data, such as air pollution. The remaining 7 risk factors were considered for the two-sample MR analyses, including blood pressure (BP) ⁷⁵, atrial fibrillation (AF) ⁷⁶, type 2 diabetes (T2D) ⁷⁷, white matter hyperintensity (WMH) ⁷⁸, body mass index (BMI) ⁷⁹, alcohol consumption and smoking behaviour ⁸⁰. ”

6. selection of top findings

currently, the top findings were selected based only on MR findings. Although a lot of MR methods had

been applied (including some new ones such as the contamination mixture method). However, I am not sure how reliable these methods are when we only have a limited number of instruments (If these methods were only applied to risk factor against stroke, then please specify this in the methods.). To me, 4 of 6 top protein-disease associations with coloc evidence should be considered as top findings.

We have now clarified that we employed the two-sample MR framework incorporating the sensitivity analyses for both primary MR (proteins → stroke outcomes), two-step MR (Step 1 MR: stroke risk factors → stroke outcomes; Step 2 MR: proteins → stroke risk factors) and Phe-MR (proteins → PheWAS). The MR methods applied in each of the MR settings depend on the number of IVs for each exposure. For example, as each of the six stroke-associated protein targets (TFPI, TMPRSS5, CD40, CD6, IL6RA, MMP12) have >3 IVs, we applied all MR methods and sensitivity analyses listed in the Methods. Detailed MR results and sensitivity analyses are provided in **Supplementary Table 3, 6, 7, 9**. We have clarified this in the **Methods (Page 21, line 491-494)** as follows:

“ We employed two-sample MR framework incorporating the sensitivity analyses for both primary MR (proteins → stroke outcomes), two-step MR (Step-1 MR: stroke risk factors → stroke outcomes; Step-2 MR: proteins → stroke risk factors) and Phe-MR (proteins → PheWAS). The MR methods applied in each of the MR settings depend on the number of IVs for each exposure. ”

We agree, where there is evidence that the pQTL and the genetic variants associated with the outcome are shared (*i.e.* colocalize) strengthens the support for the MR findings. However, we also have to acknowledge that lack of colocalization evidence does not invalidate the findings as there is a high false negative rate with colocalization methodologies (typically around 60%) [Hukku et al, 2021]. Therefore, we also consider that the two proteins that do not have strong genetic colocalization evidence with stroke outcomes, *i.e.* IL6RA and MMP12. These proteins have been associated (using MR) with other CVDs [Swerdlow et al, 2012; Sun et al, 2018] and indeed therapies that target these proteins (and their ligand/receptor) are in phase 3 clinical development for treatment of CVDs for IL6R. We have added the following text to the **Discussion (Page 13, line 321-325)**:

“ We acknowledge, where there is evidence that the pQTL and the genetic variants associated with the outcome are shared strengthens the support for the MR findings. However, we recognize that lack of colocalization evidence does not invalidate the findings as there is a high false negative rate with colocalization methodologies (typically around 60%)⁶⁰. ”

References:

Hukku, Abhay, et al. "Probabilistic colocalization of genetic variants from complex and molecular traits: promise and limitations." *The American Journal of Human Genetics* 108.1 (2021): 25-35.

Swerdlow, D.I. et al. The interleukin-6 receptor as a target for prevention of coronary heart disease: a mendelian randomisation analysis. *Lancet* **379**, 1214-24 (2012).

Sun, B.B. et al. Genomic atlas of the human plasma proteome. *Nature* 558, 73-79 (2018).

7. Coloc

The HyPrColoc method was applied for proteins vs stroke sub-types, which is interesting. However, I

feel a HyPrColoc of protein, stroke risk factors and stroke will be more meaningful. Especially for TFPI and TMPRSS5 effect on stroke. In addition, IL6RA is an important protein isoform, why not apply Susie + coloc (or PWCoCo) for this case?

Thank you for these suggestions. We have applied HyPrColoc for genetic associations across protein, stroke risk factor(s) and stroke outcomes, for 3 proteins that showed causal relationships with both risk factor(s) and stroke outcomes [Methods]. While we did see mediation effects of proteins on stroke outcomes via some risk factors, we did not see full colocalization across protein, risk factor(s), and stroke outcomes in any of the 3 protein-encoding gene regions [Supplementary Figure 2]. This might be due to the high false negative rate (typically around 60%) of statistical colocalization or lack of power in association analysis of individual traits (Hukku et al). Another potential reason is that the effects mediated by the identified risk factor(s) to stroke risk is partial (~25%) and other factors underlying the mechanism remains unknown.

We have added the following analyses to the **Methods** as follows:

“ For the three proteins that showed causal relationships with stroke risk factors, we also applied HyPrColoc across multiple traits, i.e. for TFPI, HyPrColoc was applied to TFPI pQTLs and GWAS of stroke, ischemic stroke, WMH and BMI; for TMPRSS5, HyPrColoc was applied to TMPRSS5 pQTLs and GWAS of stroke, ischemic stroke, cardioembolic stroke, and AF; for IL6R, HyPrColoc was applied to IL6R pQTLs and GWAS of stroke, ischemic stroke, cardioembolic stroke and AF. ”

We also applied PwCoCo for IL6RA and stroke outcomes to test whether the conditionally independent signal(s) of IL6RA pQTLs were shared with stroke associations. However, we did not find evidence of colocalization of secondary or tertiary IL6RA pQTLs and stroke associations. PwCoCo results are provided in the table below.

Colocalization results of IL6RA pQTLs and stroke associations using PwCoCo.

pQTL	Stroke outcomes	SNP1 (pQTL)	SNP2 (stroke signal)	H0	H1	H2	H3	H4	log_abf_all
	Stroke	unconditioned	unconditioned	0	0.682	0	0.250	0.068	2684.2
		chr1:154382049	unconditioned	0	0.688	0	0.244	0.069	2685.6
		chr1:154408916	unconditioned	0	0.687	0	0.245	0.069	2685.6
		chr1:154426264	unconditioned	0	0.687	0	0.245	0.069	1192.26
	Ischemic-stroke	unconditioned	unconditioned	0	0.682	0	0.250	0.068	2684.2
		chr1:154382049	unconditioned	0	0.688	0	0.243	0.069	2685.6
		chr1:154408916	unconditioned	0	0.687	0	0.245	0.069	2685.6
		chr1:154426264	unconditioned	0	0.687	0	0.245	0.069	1192.26
IL6RA	Large-artery-stroke	unconditioned	unconditioned	0	0.681	0	0.251	0.068	2684.2
		chr1:154382049	unconditioned	0	0.687	0	0.244	0.069	2685.6
		chr1:154408916	unconditioned	0	0.686	0	0.245	0.069	2685.6
		chr1:154426264	unconditioned	0	0.686	0	0.245	0.069	1192.26
	Cardioembolic-stroke	unconditioned	unconditioned	0	0.682	0	0.250	0.068	2684.2
		chr1:154382049	unconditioned	0	0.687	0	0.244	0.069	2685.6
		chr1:154408916	unconditioned	0	0.686	0	0.245	0.069	2685.6
		chr1:154426264	unconditioned	0	0.686	0	0.245	0.069	1192.26
	Small-vessel-stroke	unconditioned	unconditioned	0	0.682	0	0.250	0.068	2684.2
		chr1:154382049	unconditioned	0	0.687	0	0.244	0.069	2685.38
		chr1:154408916	unconditioned	0	0.686	0	0.245	0.069	2685.38
		chr1:154426264	unconditioned	0	0.686	0	0.245	0.069	1192.06

M

Supplementary Figure

2. Colocalization plots of pQTLs, and genetic associations of stroke outcomes and risk factor(s).

M. Stacked genetic association plots of TFPI pQTLs, Stroke, Ischemic-stroke, WHM and BMI associations.

N. Stacked genetic association plots of TMPRSS5 pQTLs, Stroke, Ischemic-stroke, Cardioembolic-stroke and AF associations.

O. Stacked genetic association plots of IL6RA pQTLs, Stroke, Ischemic-stroke, Cardioembolic-stroke and AF associations.

N

O

8. mediator (optional)

The current setting testing protein  risk factor  stroke. However, some recent studies showed that BMI causally influence protein levels (e.g. CRP). Will it be useful to also consider risk factor  protein -> stroke. This may further increase the complexity of the study design. So just a suggestion.

In the current study, the risk factor -> protein relationships were only considered for the top six protein targets to test whether there was evidence of reverse causation. We found there was no reverse causation for any of the six protein targets to stroke outcomes. We agree, this would be a valuable analysis to provide, but share the reviewers concern that it could add complexity to the paper with potential to confuse the reader, so have not investigated the mediation analysis with protein as mediator for all the proteins.

Minor comments

Line 72, Put a reference if there is such a tissue specific eQTL study on IBD?

We have now added the following reference:

Ref: Hular, Imge, et al. "Enrichment of inflammatory bowel disease and colorectal cancer risk variants in colon expression quantitative trait loci." BMC genomics 16.1 (2015): 1-15.

Line 74, some more proteome GWAS coming out recently in Science and Nature Genetics.

We now reference these two proteome GWAS which were published while our paper was under review.

Ref1: Ferkingstad, Egil, et al. "Large-scale integration of the plasma proteome with genetics and disease." Nature genetics 53.12 (2021): 1712-1721.

Ref2: Pietzner, Maik, et al. "Mapping the proteo-genomic convergence of human diseases." Science 374.6569 (2021): eabj1541.

Line 80, a reference for MR vs RCT

We have now added:

Ref: Davey Smith, George, and Shah Ebrahim. "'Mendelian randomization': can genetic epidemiology contribute to understanding environmental determinants of disease?" International journal of epidemiology 32.1 (2003): 1-22.

Line 129, why season was controlled as a covariate?

Season is used as a covariate as it is a determinant of the levels of some proteins so accounting for this non-genetic source of variation will improve power to detect genetic association signals. We have added the following text to the **Methods (page 17, line 398-400)** and include the following reference that shows seasonal variation in protein levels.

" Season is used as a covariate as it is a determinant of the levels of some proteins ⁶⁶ so accounting for this non-genetic source of variation will improve power to detect genetic association signals. "

Reference: Enroth, Stefan, et al. "Effects of long-term storage time and original sampling month on biobank plasma protein concentrations." *EBioMedicine* 12 (2016): 309-314.

Line 268, "causal associations" changes to "causal relationships"

Changed as suggested.

Line 275 (and other places), not a big fan of define "significance" based on p value.

We agree that p-value is not perfect for differentiating significant findings when many tests are performed, however the reader would likely have a good sense of how to interpret the findings where we claim significant associations where we provide a P-value threshold for guidance as this is still common practice. We have now also provided FDR in **Supplementary Table 3** to further help with interpretability. We perform additional sensitivity analyses to validate the "significant" findings to minimise false positive. We have also minimised the use of the word significant.

Line 296, as mentioned above, cis acting pQTLs could be pleiotropic too. need to do a test.

Agree. Please refer to our response to your earlier comment #3.

Line 363, very interesting finding, can do a mediation analysis to estimate the % of mediation effect.

Thank you for the suggestion. We have now added the % of mediation effect analysis to the DAG plots in **Figure 6** and to the **Results (page 9, line 203-213)**.

“ Mediation effect of proteins on stroke outcomes via risk factors

To investigate the indirect effect of proteins on stroke outcomes via risk factors, we carried out mediation analysis using the effect estimates from two-step MR and the total effect from primary MR. This analysis was restricted to three proteins, *i.e.* TFPI, TMPRSS5, and IL6RA, that showed evidence of an effect in the MR analyses with risk factors and stroke outcomes. We used the product method to estimate the indirect effect and the delta method to estimate the standard errors (SE) and confidential interval (CI) (**Methods**). The proportion of mediation effect of TFPI via WMH is about one fifth (20.8%), while the mediation effect via BMI is modest (3.8%). The indirect effect of TMPRSS5 on risk of cardio-embolic stroke via AF contributes to a quarter of the total effect (24.7%). Similarly, the proportion of mediation effect of IL6RA on stroke via AF is 27.6% (**Fig. 6 and Supplementary Table 15**). ”

Figure 3 & 4, put number of SNPs into the forest plots?

The number of SNPs have now been added to the forest plots.

Figure 6, "A. Stroke" Is this a typo?

Yes. Now corrected.

btw, please change the manuscript to a N.C. format, e.g. abstract and order of sections.

We have now changed the manuscript to the N.C. format.

Reviewer #2 (Remarks to the Author):

Chen et al have investigated whether genetically determined blood levels of over 300 proteins are associated with ischemic stroke. Significant associations may be indicative of causal effects. They identify 6 proteins that are robustly associated to ischemic stroke. These are important results from a well-executed study, expanding our understanding of stroke etiology and providing new potential drug targets.

Response

We thank the reviewer for the positive comments and suggestions. We have updated the manuscript to clarify the story and add the details the reviewer requested. Furthermore, we have performed additional analyses as appropriate. We thank the reviewer for helping us to improve the clarity of the manuscript.

Major comments

Multiplex platforms such as O-link are not gold standard measure of protein levels. The authors do note that O-link cannot distinguish between free or bound protein nor active or inactive, but there may be differences beyond that just based on the technology. Could there be cis-eQTLs that influence binding of the O-link assay as opposed to protein levels? If so, would that create false positive MR associations or false negative MR associations? Could the authors discuss (in their limitations section) this topic in some more depth?

Thank you very much for the very good points.

We acknowledge there are limitations regarding the current multiplex platforms. But cis-eQTLs are less likely to influence the protein abundances as measured by antibody-based Olink assays comparing to the Somalogic aptamer-based assays [Pietzner et al]. In terms of the protein instruments we used in the current study, IV(s) for all proteins but MMP12, are significant eQTLs in at least one tissue from GTEx V8. The effect direction of these variants on gene expression and protein levels are consistent for TFPI, CD40, CD6 and TMPRSS5, but not for IL6R. The diverse effect direction of variants on *IL6R* gene expression and IL6RA protein level is likely due to the measurement of IL6RA. The O-link assay only captures the circulating free IL6RA while all isoforms of *IL6R* transcripts are captured collectively by the gene expression measurements in tissues (blood and artery).

According to Ferkingstad et al's recent study of proteome on Somalogic aptamer-based platform, if a pQTL is in high LD with cis-eQTL, it is less likely to lead to binding artefacts. However, we are aware that some cis-eQTLs are due to effects on splicing and isoforms rather than RNA quantity which might have a direct impact on the antibody binding and would have to be understood on a case by case basis.

We have added the following text to the **Discussion (page 14, line 335-350)**:

“ We acknowledge there are limitations regarding the current multiplex platform for protein abundance measurements. For instance, IVs linking to a protein altering variant (PAV) can influence the measurement of the protein binding affinity ('abundance'), leading to

disconnection of the protein abundance and the function of the mutant protein⁶³. But cis-eQTLs are less likely to influence the protein abundances as measured by antibody-based Olink assays comparing to the Somalogic aptamer-based assays⁶³. A recent proteome study¹⁵ using Somalogic aptamer-based platform showed that cis-eQTLs were less likely to affect the protein binding. Indeed, if a significant pQTL was in high LD with cis-eQTL, it was less likely to lead to binding artefacts. To test whether the IV(s) for the 6 target proteins were cis-eQTLs for their protein-encoding gene, we sought eQTLs from GTEx V8 and identified that IV(s) for 5 of the proteins (not MMP12) were significant eQTLs in at least one tissue. The effect direction of these variants on gene expression and protein levels were consistent for TFPI, CD40, CD6 and TMPRSS5, except for IL6R. The diverse effect direction of variant on *IL6R* gene expression and IL6RA protein level is likely due to the measurement of IL6RA. The O-link assay only captures the circulating free IL6RA while all isoforms of *IL6R* transcripts are captured collectively by the gene expression measurements in tissues (blood and artery).

”

Reference:

Pietzner, Maik, et al. "Synergistic insights into human health from aptamer-and antibody-based proteomic profiling." *Nature communications* 12.1 (2021): 1-13.

Ferkingstad, Egil, et al. "Large-scale integration of the plasma proteome with genetics and disease." *Nature genetics* 53.12 (2021): 1712-1721.

Suggest being more transparent about the number of instrumental variables that are behind the core 6 MR associations. If there are more than 3 instruments it would be nice to see the median-based, mode-based, and Egger-based MRs. Although MR-Egger was used according to the flowchart, I was not able to find references to it in the main text (except for references to the MR-Egger intercept).

Thank you for the suggestions. We have added the number of instrumental variables for the 6 proteins that are associated with stroke outcome(s) to the forest plot presented in **Figure 3**. We have applied MR-Egger and MR Contamination mixture model for causal effect estimation for proteins with >3 IVs. The Egger method is added to **Methods**. Full details of MR results derived from MR-IVW, MR-Egger (not just intercept information), and MR-ConMix are provided in **Supplementary Table 3**. MR causal estimates derived from median-based and mode-based methods as suggested are also reported in **Supplementary table 3**.

Minor comments

“The association of TFPI, IL6RA and TMPRSS5 with stroke could be mediated by these risk factors, such as body mass index, white matter hyperintensity and atrial fibrillation.” This sentence sounds a bit

vague, as if you are stating a hypothesis instead of a result. Replace “could be” with something like “appeared to be”?

Revised as the reviewer suggests (line 52).

Does it make sense to use $5E-8$ as the significance threshold in the identification of pQTLs, when 354 proteins were tested?

Please refer to reply to reviewer 1 comment #4.

The “phenome-wide MR” section in the methods focused on 6 stroke associated proteins currently comes before the section that describes the identification of these 6 proteins. Suggest moving it further back.

We have moved it as you have suggested.

For the phenome-wide MR section of the results, you note that “If the effect direction of the protein on the disease or trait is the same as on stroke, the effect is considered “beneficial” and “deleterious” otherwise.” For clarify I suggest using this beneficial/deleterious terminology in the next sentence (and throughout) as well.

Thank you for the suggestion, we have changed this throughout the text.

REVIEWER COMMENTS

Reviewer #1 (Remarks to the Author):

Thanks for all the efforts from Chen et al. The authors took good care of most of my questions. The readership of the manuscript and robustness of the findings have been improved. I have few additional suggestions, which hopefully will further improve the robustness of the findings.

1. Figure 1. Nice summary of this comprehensive study. Thanks. Three suggestions:

1.1 For step 3 (MR of protein on risk factors), I guess HyPrColoc was applied, please add.

1.2 Has colocalization been conducted in step 5 (PheMR)? Chris Wallace group recently developed a PheWAS version of coloc, which could be a good fit for step 5.

1.3 Any approach to reformat it into two rows/columns so the font size is larger?

2. Figure 6. Thank you for the nice plot to summarise the top findings. Since TFPI effect on any stroke was mediated by BMI and WMH. I am wondering whether the WMH effect will be influenced by BMI? Will a MVMR model be useful here?

3. novelty of the findings. Thanks for conducting a comparison with INTERVAL results, which increased the reliability of the findings. Also worth validating the findings with more recent version of SOMAlogic data, e.g. ARIC/Fenland/Decode. Recommend a recent proteome study using ARIC data, where stroke is one of the outcomes been included.

<https://www.medrxiv.org/content/10.1101/2022.01.09.21268473v1>

4. mediation analysis, I would like to check with the authors that sample overlap is not a major issue here.

5. colocalization of top findings. Thank you for providing the additional information. The regional plots are useful to clarify things. By checking the regional plots, I agree with the authors that some of them showed good evidence but the top hit was hidden so coloc can not detect the shared causal effects. The IL6R on stroke finding could be a power issue.

For those proteins with less coloc evidence, I suggest to down weight them as secondary findings. or give a warning in discussion.

Reviewer #2 (Remarks to the Author):

No further comments: the authors thoroughly addressed the comments from both reviewers.

Response to REVIEWER COMMENTS

Reviewer #1 (Remarks to the Author):

Thanks for all the efforts from Chen et al. The authors took good care of most of my questions. The readership of the manuscript and robustness of the findings have been improved. I have few additional suggestions, which hopefully will further improve the robustness of the findings.

We appreciate the reviewer's comments and suggestions. We have updated the manuscript and added the details the reviewer requested. We thank the reviewer for helping us to improve the clarity and robustness of the manuscript.

1. Figure 1. Nice summary of this comprehensive study. Thanks. Three suggestions:

1.1 For step 3 (MR of protein on risk factors), I guess HyPrColoc was applied, please add.

Response to 1.1: Yes, we have applied HyPrColoc for pQTLs and risk factors for Step 3 and provided the results in Supplementary Table 4-(3). We have added this to **Figure 1**.

1.2 Has colocalization been conducted in step 5 (PheMR)? Chris Wallace group recently developed a PheWAS version of coloc, which could be a good fit for step 5.

Response to 1.2: Thank you very much for the suggestions. We have updated the analyses and applied colocalization - HyPrColoc for Step 5. This is also added to **Figure 1** and Supplementary Table 4-(4).

1.3 Any approach to reformat it into two rows/columns so the font size is larger?

Response to 1.3: We have now increased the font size used in **Figure 1** to make it more readable.

Fig. 1 | Overview of this MR study.

Four O-link panels were used to measure plasma proteins in a subset of ~ 5000 samples from the INTERVAL study. Genetic variants associated with plasma protein levels were identified based on results from their corresponding GWAS. These genetic variants were then used as proxies for the protein level and tested their relationship with stroke used data from the MEGASTROKE consortium for stroke outcomes (Primary MR), and with conventional stroke risk factors that were causally associated with stroke (Secondary MR). Colocalization analyses were performed to test the shared genetic associations of protein level, stroke outcomes and risk factors. Mediation analyses by two-step MR were performed for proteins that potentially causally associated with both risk factors and stroke outcomes. We also tested the relationships of the potentially causal stroke proteins with 784 phenotypes in UK Biobank to test a broad spectrum of potential effects of hypothetical therapeutic agents for stroke. #Stroke outcomes: Any Stroke; Any Ischemic Stroke; Large Artery Stroke; Cardioembolic Stroke; Small Vessel Stroke.

2. Figure 6. Thank you for the nice plot to summarise the top findings. Since TFPI effect on any stroke was mediated by BMI and WMH. I am wondering whether the WMH effect will be influenced by BMI? Will a MVMR model be useful here?

Response to 2: We have investigated this in two ways because we were concerned that the MVMR would be biased towards BMI due to the larger number of IVs available for BMI compared to WMH. First, we performed bi-directional MR for BMI \leftrightarrow WMH and found that in either direction there are no significant associations between these two factors (Table A). Secondly, we considered MVMR and found that only BMI showed significant associations with Stroke in both models: 1) Stroke \sim BMI + WMH (Table B1) and 2) Stroke \sim TFPI + BMI + WMH (Table B2), which showed that adjusting for the effects of WMH, BMI still showed causal associations with stroke. Additionally, it typically isn't feasible to conduct an MVMR analysis using variants from a single gene region, as such an analysis requires multiple independent genetic predictors of the protein in that region (one more independent genetic predictor that there are mediators is required for estimation to be possible, more are necessary for reliable estimation).

Table A. Bi-directional MR: BMI and WMH

Exposure	Outcome	SNPset	NSNPs	MR_IVW			MR_PRESSO		
				beta	se	Pvalue	beta	se	Pvalue
BMI	WMH	Raw	780	0.031	0.023	0.164	0.031	0.023	0.164
		Outlier-corrected*	774	0.027	0.021	0.196	0.027	0.021	0.197
WMH	BMI	Raw	16	0.021	0.033	0.523	0.021	0.033	0.533
		Outlier-corrected	11	-0.011	0.015	0.468	-0.011	0.015	0.484

*Outlier-corrected indicates the outliers that identified by MR-PRESSO have been removed in the MR analysis.

Table B1. Multivariable MR: Stroke \sim BMI + WMH

Outcome	NSNPs	Exposures	OR [95%CI]	P-value	I^2
Stroke	813	BMI	1.19 [1.12, 1.25]	6.82E-10	27%
		WMH	1.03 [0.963, 1.1]	0.405	
Ischemic-stroke	813	BMI	1.18 [1.11, 1.25]	4.83E-08	26%
		WMH	1.04 [0.965, 1.11]	0.335	
Large-artery-stroke	813	BMI	1.29 [1.12, 1.49]	0.000322	19%
		WMH	1.05 [0.885, 1.24]	0.599	
Cardioembolic-stroke	813	BMI	1.15 [1.03, 1.27]	0.00942	13%
		WMH	1.08 [0.95, 1.22]	0.245	
Small-vessel-stroke	813	BMI	1.12 [0.981, 1.27]	0.0943	18%
		WMH	1.14 [0.977, 1.33]	0.0971	

Table B2. Multivariable MR: Stroke \sim TFPI + BMI + WMH

Outcome	NSNPs	Exposures	OR [95%CI]	P-value	I^2
Stroke	830	TFPI	0.99 [0.969, 1.01]	0.384	27%
		BMI	1.19 [1.13, 1.26]	3.31E-10	
		WMH	1.04 [0.976, 1.11]	0.223	
Ischemic-stroke	830	TFPI	0.992 [0.969, 1.02]	0.531	27%
		BMI	1.18 [1.12, 1.26]	2.58E-08	
		WMH	1.05 [0.978, 1.13]	0.176	
Large-artery-stroke	830	TFPI	0.988 [0.934, 1.05]	0.668	19%
		BMI	1.29 [1.12, 1.49]	0.000304	
		WMH	1.06 [0.898, 1.25]	0.484	
Cardioembolic-stroke	830	TFPI	0.992 [0.951, 1.03]	0.713	13%
		BMI	1.16 [1.04, 1.28]	0.00627	
		WMH	1.08 [0.955, 1.23]	0.212	
Small-vessel-stroke	830	TFPI	0.989 [0.939, 1.04]	0.69	18%
		BMI	1.12 [0.98, 1.27]	0.0982	
		WMH	1.15 [0.989, 1.34]	0.0684	

3. novelty of the findings. Thanks for conducting a comparison with INTERVAL results, which increased the reliability of the findings. Also worth validating the findings with more recent version of SOMALogic data, e.g. ARIC/Fenland/Decode. Recommend a recent proteome study using ARIC data, where stroke is one of the outcomes been included.

<https://www.medrxiv.org/content/10.1101/2022.01.09.21268473v1>

Response to 3: Thank you for suggestions. We have looked into the MR analysis using ARIC data and GBMI Stroke GWAS in Zhao et al's study as suggested. However, the authors do not identify a causal relationship between TFPI or TMPRSS5 and stroke, which is not unexpected due to the lack of statistical power of the ARIC study to identify stroke associated proteins. The ARIC paper used just 15,842 cases which is considerably less than the number available in MEGASTROKE, 40,585,. We note that the MEGASTROKE study tripled the number of stroke-associated loci compared to early stroke GWAS (which were of comparable size to the stroke data used in the ARIC study) clearly illustrating the value of the substantially larger study from MEGASTROKE that we have used. Furthermore, the stroke phenotype used by ARIC is restricted to any stroke, while in our study we show the strongest association of TFPI is with ischemic stroke and of TMPRSS5 is with cardioembolic stroke.

4. mediation analysis, I would like to check with the authors that sample overlap is not a major issue here.

Response to 4: Sample overlap is not an issue here, because we used proteome data from INTERVAL cohort, which is an independent cohort from the outcome GWASs, e.g. MEGASTROKE for stroke GWAS and stroke risk factor GWAS from various consortia (**Table 1**). While there is no overlap between the sample for the protein exposures and for the disease outcome, there is some overlap between the samples used for the proposed mediators and for the outcome. However, the extent of overlap is not great, and so bias is not expected to be substantial ^{Ref}.

Reference: Burgess, S., Davies, N.M. and Thompson, S.G., 2016. Bias due to participant overlap in two-sample Mendelian randomization. Genetic epidemiology, 40(7), pp.597-608.

5. Colocalization of top findings. Thank you for providing the additional information. The regional plots are useful to clarify things. By checking the regional plots, I agree with the authors that some of them showed good evidence but the top hit was hidden so coloc cannot detect the shared causal effects. The IL6R on stroke finding could be a power issue. For those proteins with less coloc evidence, I suggest to down weight them as secondary findings. Or give a warning in discussion.

Response to 5: We agree with the reviewer and therefore have provided the suggested warning in the **Discussion (Page 13, line 317-321)**:

“ We acknowledge, where there is evidence that the pQTL and the genetic variants associated with the outcome are shared, strengthens the support for the MR findings. However, we recognize that lack of colocalization evidence does not invalidate the findings

as there is a high false negative rate with colocalization methodologies (typically around 60%)⁶⁰. ”

60. Hukku, A., Pividori, M., Luca, F., Pique-Regi, R., Im, H.K. and Wen, X., 2021. Probabilistic colocalization of genetic variants from complex and molecular traits: promise and limitations. *The American Journal of Human Genetics*, 108(1), pp.25-35.

Reviewer #2 (Remarks to the Author):

No further comments: the authors thoroughly addressed the comments from both reviewers.

REVIEWERS' COMMENTS

Reviewer #1 (Remarks to the Author):

Thank you for all the efforts from Chen et al during this revision stage.

They have answered all my questions. The paper is in a good quality. I have no further comments.